# An immortal porcine preadipocyte cell strain for efficient production of cell-cultured fat

Yun-Mou Cheng[1,2,3], Peng-Cheng Hong[1,2,3], Ming-Mei Song[1,2], Hai-Ning Zhu[1,2], Jing Qin[1,2], Zeng-Di Zhang [1,2], Hao Chen[1,2], Xing-Zhou Ma[1,2], Meng-Yuan Tian[1,2], Wei-Yun Zhu[1,2] & Zan Huang [1,2✉]

Adding adipose cells to cell-cultured meat can provide a distinctive aroma and juicy texture similar to real meat. However, a significant challenge still exists in obtaining seed cells that can be propagated for long periods, maintain their adipogenic potential, and reduce production costs. In this study, we present a cell strain derived from immortalized porcine preadipocytes that can be subculture for over 40 passages without losing differentiation capacity. This cell strain can be differentiated within 3D bioscaffolds to generate cell-cultured fat using fewer chemicals and less serum. Additionally, it can be expanded and differentiated on microcarriers with upscaled culture to reduce costs and labor. Moreover, it can co-differentiate with muscle precursor cells, producing a pattern similar to real meat. Therefore, our cell strain provides an exceptional model for studying and producing cell-cultured fat.

[1] Laboratory of Gastrointestinal Microbiology, Jiangsu Key Laboratory of Gastrointestinal Nutrition and Animal Health, College of Animal Science and Technology, Nanjing Agricultural University, Nanjing, China. [2] National Center for International Research on Animal Gut Nutrition, Nanjing Agricultural University, Nanjing, China. [3] These authors contributed equally: Yun-Mou Cheng, Peng-Cheng Hong. ✉email: huangzan@njau.edu.cn

The global demand for meat is growing due to population growth and increasing living standards. However, traditional meat production methods are associated with various environmental problems. Cellular agriculture is an alternative method that uses biological techniques to produce cultured meat from meat cells grown in vitro. This method significantly reduces cultivation time, energy consumption, and pollutant emissions, making it a sustainable solution to traditional extensive husbandry production[1].

While most cell-cultured meat research focuses on muscle cells and tissues, fat is also a crucial component of meat. Fat provides essential nutrients and contributes to the meat's taste, texture, and tenderness, which significantly influences customer preferences[2–4]. Accordingly, vegetable oil is added to cultured meat to mimic real meat's sensory and mechanical properties. However, the carbon chain length and saturation of vegetable oil differ significantly from those of real meat[5,6]. Moreover, meat fat is wrapped in the cells by multilayer cellular membranes, resulting in different droplet sizes, distribution, and antioxidant capacity of lipids compared to vegetable oil. Therefore, adding in vitro cultured fat cells is more suitable for improving the flavor and texture of cell-cultured meat.

One of the main challenges of cell-cultured fat is to obtain a stably proliferative and highly adipogenic seed cell. Adipocytes have terminally differentiated and have lost their ability to divide. Hence, several primary adipogenic cells are used as seed cells in cultured meat research[7]. However, their adipogenic efficiency and proliferation potency decrease with the increasing number of passages, limiting their utility for cell-cultured fat research and large-scale production.

To overcome these issues, researchers have immortalized preadipocytes to maintain their proliferative and differentiation capacity. Most of these cell lines are from rodents and avians, such as 3T3-L1 and IPC1[8,9]. Several porcine cell lines with adipogenic potential have also been reported that can maintain proliferation over 30 passages[10–13]. However, all cell lines have relatively low adipogenic efficiency, and none of them have been proven or available for using in cell-cultured fat. Therefore, a highly adipogenic immortal porcine cell line with cultured fat potency is demanded in cell-cultured meat research and production.

In this study, we reported a highly adipogenic immortal porcine cell strain, ISP-4 (Immortalized Swine Preadipocytes #4), with ideal characteristics for cultured fat research and production. ISP-4 could be continuously passaged for more than 40 generations without attenuating adipogenic efficiency, and its adipogenic process can be achieved with less serum or fewer food toxic chemicals in the differentiation cocktail. Moreover, ISP-4 could be differentiated into mature fat cells within a bioscaffold to form cell-cultured meat, or co-cultured with myoblast cells, and simultaneously differentiated into mature adipocytes and muscle tubes. We also showed that ISP-4 could be amplified and differentiated into adipocytes on edible microbeads in the suspension culture system. These findings suggest that ISP-4 is an idea model for cell-cultured fat, and has considerable potential for cell-cultured meat production.

## Results

### An immortal porcine preadipocyte cell line with high differentiation potential

In a previous study, we isolated 108 cell strains from SV40T immortalized porcine adipocyte cells using the limiting dilution method and found that five of them could accumulate lipids after differentiation[14]. Among these, ISP-4 demonstrated the highest adipogenic potency and was selected for further investigation in cultured fat research (Supplementary Fig. 1). To optimize its growth rate for large-scale cell-cultured fat production, we first tested the effect of culture medium on ISP-4 growth. Our results showed that DMEM slightly increased the cell growth rate compared to the DMEM/F12 medium, possibly due to the rapid acidification of DMEM/F12 (Supplementary Fig. 2a–c). Subsequently, we examined the adipogenic efficiency of ISP-4 in both media and found that it accumulated more lipid droplets in DMEM (Supplementary Fig. 2d), indicating that DMEM is a more suitable medium for the ISP-4 cell strain.

While the adipogenesis ability of primary porcine preadipocytes declines with increasing passage number, we investigated whether ISP-4 could maintain its adipogenesis efficiency when cultured in DMEM. To date, ISP-4 has been subcultured for over 40 generations, and its growth rate has not significantly decreased (Fig. 1a). Moreover, we found that early and late passage ISP-4 cells accumulated a comparable amount of lipids (Fig. 1b, c, e, f), and the expression of adipose-specific genes remained unchanged (Fig. 1d, g). These results suggest that ISP-4 can maintain its adipogenic potential even at higher passage numbers in DMEM.

### Optimizing adipogenic methods for ISP-4

Typically, the classical "4 + 4" protocol is used to induce adipogenesis in preadipocytes, which involves treating the cells with adipogenic differentiation medium (ADM) for 2-4 days, followed by maintenance in AMM for another 4–6 days until lipid droplets are observed[15]. However, the chemicals in ADM, such as 3-isobutyl-methylxanthine (IBMX), indomethacin, dexamethasone, rosiglitazone, can be expensive and toxic if overdosed. To address this issue, we developed a "2 × 5" protocol, where ISP-4 was differentiated in AMM only and the medium was changed every other day (Fig. 2a). After a period of 10 days, ISP-4 accumulated a substantial amount of lipids, though not as much as with the standard 4 + 4 protocol (Fig. 2b, c). Importantly, this lipid accumulation is not the result of the highly adipogenic potential of ISP-4. This was demonstrated as ISP-4 was unable to accumulate lipids when cultured without insulin for 10 days, parallel to the 2 × 5 protocol (Supplementary Fig. 3).

Serum is the most expensive component of cell culture medium. Therefore, reducing or eliminating serum in the medium will significantly improve the cost advantage of culture meat[16]. Accordingly, we investigated whether ISP-4 could differentiate under a lower serum condition. ISP-4 was differentiated with the 4 + 4 protocol, but serum was reduced to 2%. The lipid staining showed that compared to 10% FBS, ISP-4 accumulates a similar amount of lipids in 2% FBS medium (Fig. 2b, c). These results demonstrate that ISP-4 is a preadipocyte cell strain with enhanced food safety and cost advantage for the culture meat industry.

### ISP-4 can differentiate into mature adipocytes in 3-dimensional bioscaffolds

The cultivation and differentiation of seed cells on edible bioscaffolds are necessary steps in the production of cultured meat[17–19]. Therefore, we investigated whether ISP-4, which was able to differentiate into mature adipocytes in 2D cultures, could grow and differentiate in the context of 3D tissue biomimetics. Alginate, a natural polymer commonly used in the food industry and cell-cultured fat, was mixed with ISP-4 and formed into microfibers with $CaCl_2$ solutions. After 4 days of recovery in the growth medium, the microfibers were differentiated by using the 4 + 4 adipogenic protocol (Fig. 3a). The resulting ISP-4 cultured fat dramatically differed from the empty alginate hydrogel in appearance, the yellow hue of ISP-4 cultured fat is similar to other cultured fat with primary cells reported before[18] (Fig. 3b).

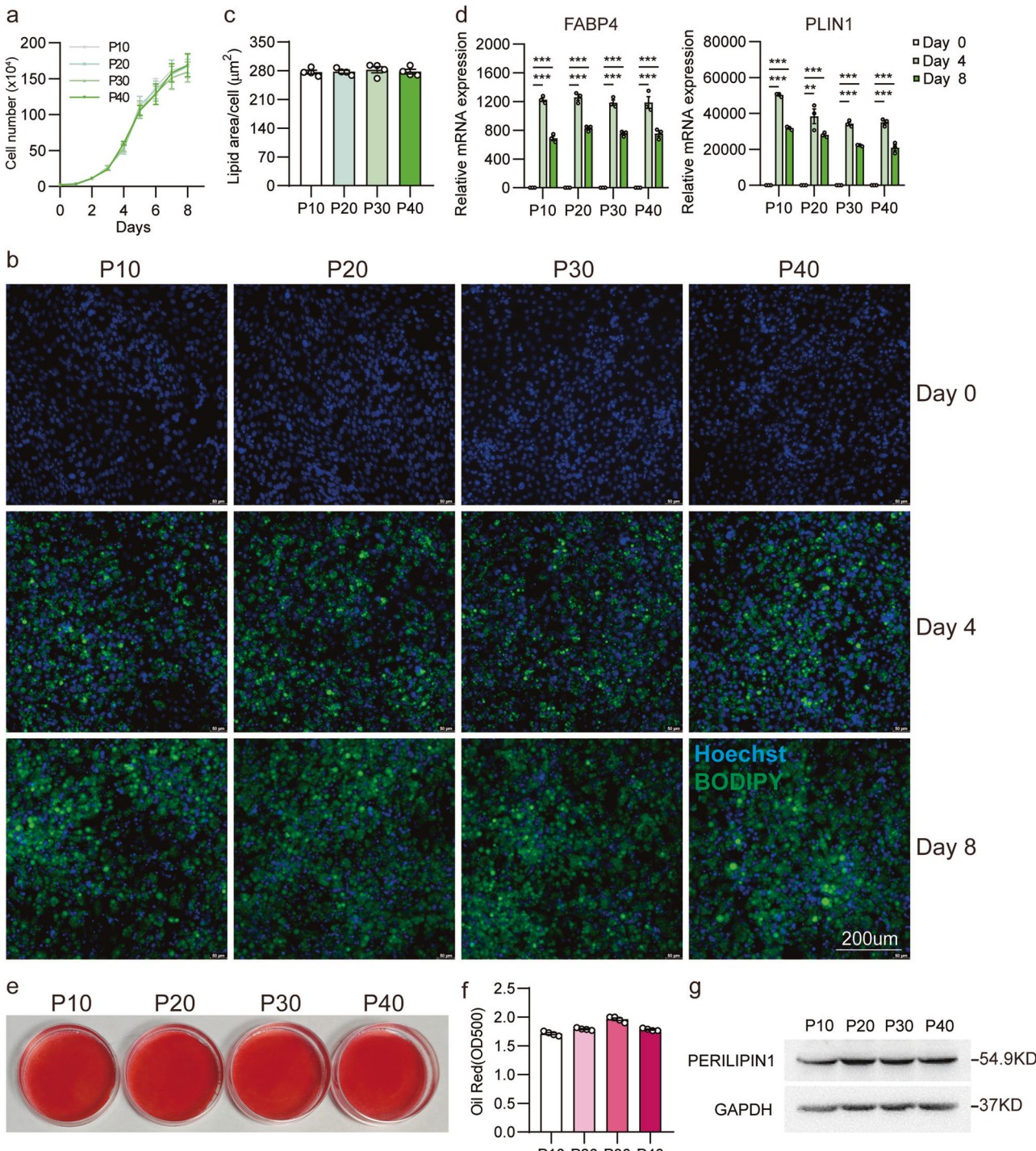

**Fig. 1 ISP-4 preserves adipogenic capability during long-term culture. a** Growth curves of ISP-4 at indicated passages ($n = 3$ biologically independent samples). **b** Representative fluorescent images of differentiated ISP-4 at indicated passages. Green = BODIPY, lipid, blue = Hoechst, nuclei. Scale bar = 200 μm. **c** The lipid content per cell was quantified by measuring the BODIPY and Hoechst fluorescence area in figure **a**, data were collected by ImageJ ($n = 4$ biologically independent samples). **d** The expression level of adipocyte-specific genes in ISP-4 at indicated passages on day 0, 4, 8 of adipogenic differentiation, measured by RT-qPCR and normalized against Day 0 ($n = 4$ biologically independent samples). **e** Oil-Red staining of differentiated ISP-4 at indicated passages. **f** Lipid contents in (**e**). were measured at 500 nm with a microplate reader ($n = 4$ biologically independent samples). **g** Immunoblotting against PLIN1 in differentiated ISP-4, GAPDH was used as loading control. All numerical values are presented as mean ± SEM; Two-tailed unpaired Student's $t$-test $p$-values are indicated as: $^*P \leq 0.05$; $^{**}P \leq 0.01$; $^{***}P \leq 0.001$.

The structure of the ISP-4 cultured fat was analyzed using BODIPY and H&E staining. After 4 days of culturing in the growth medium, only a few cells contained small lipid droplets. Interestingly, most cavities in the alginate were occupied by 2–3 cells, suggesting that ISP-4 may still divided in the alginate hydrogels before adipogenesis (Fig. 3c). This hypothesis is supported by the fact that most cells were Ki67[+] on day 0 (Supplementary Fig. 4). After four days of ADM treatment, a

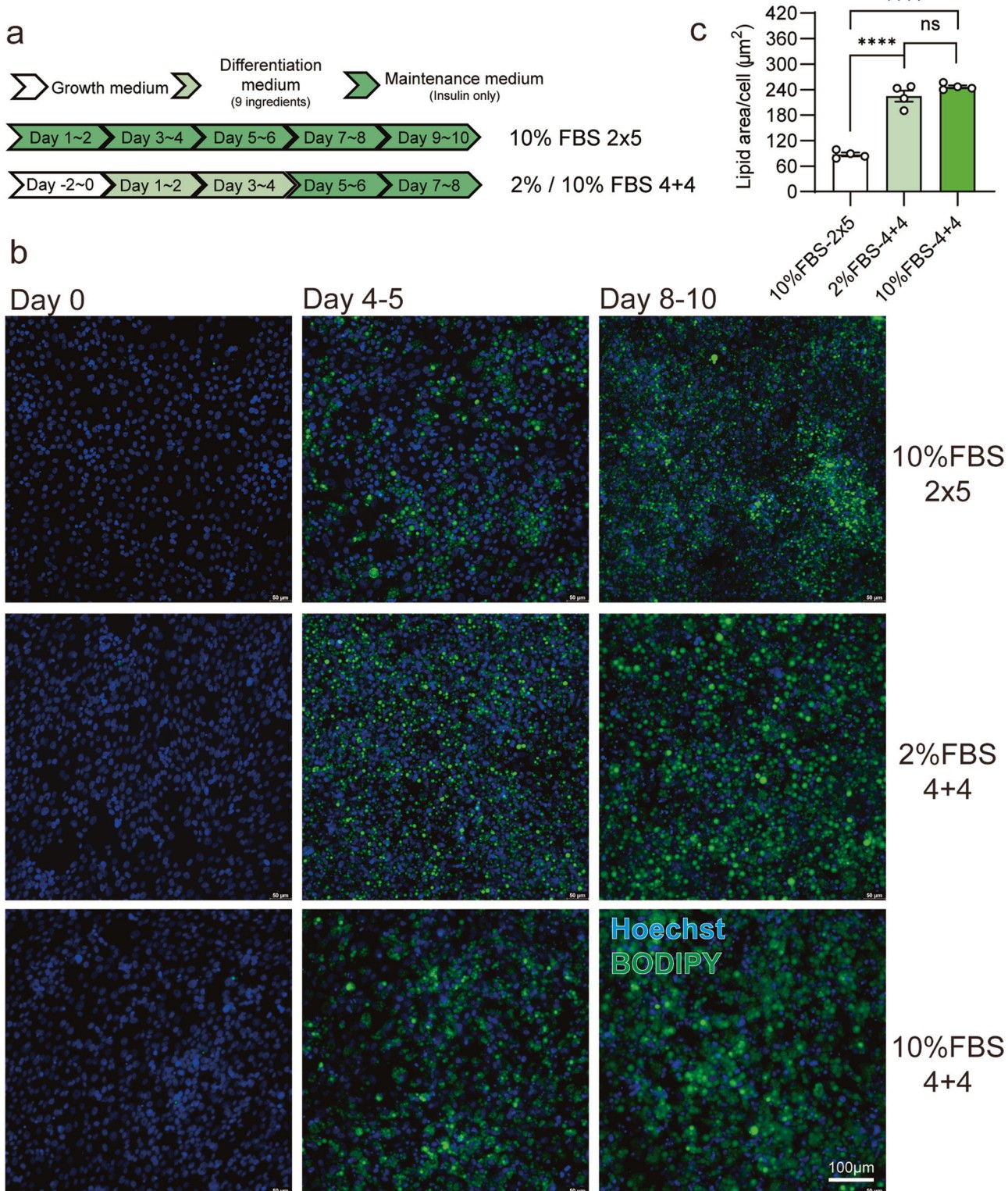

**Fig. 2 Optimization of adipogenic methods for ISP-4 in 2D culture. a** Overview of workflow for adipogenic differentiation methods of ISP-4.
**b** Representative fluorescent images of ISP-4 at indicated time point during adipogenic differentiation with different methods. Green = BODIPY,
blue = Hoechst, scale bar = 100 μm. **c** The lipid content per cell was quantified by measuring the BODIPY and Hoechst fluorescence area in figure **b**, data
were collected by ImageJ ($n = 4$ biologically independent samples). All numerical values are presented as mean ± SEM. Two-tailed unpaired Student's $t$-
test $p$-values are indicated as ***$P \leq 0.001$, ****$P \leq 0.0001$.

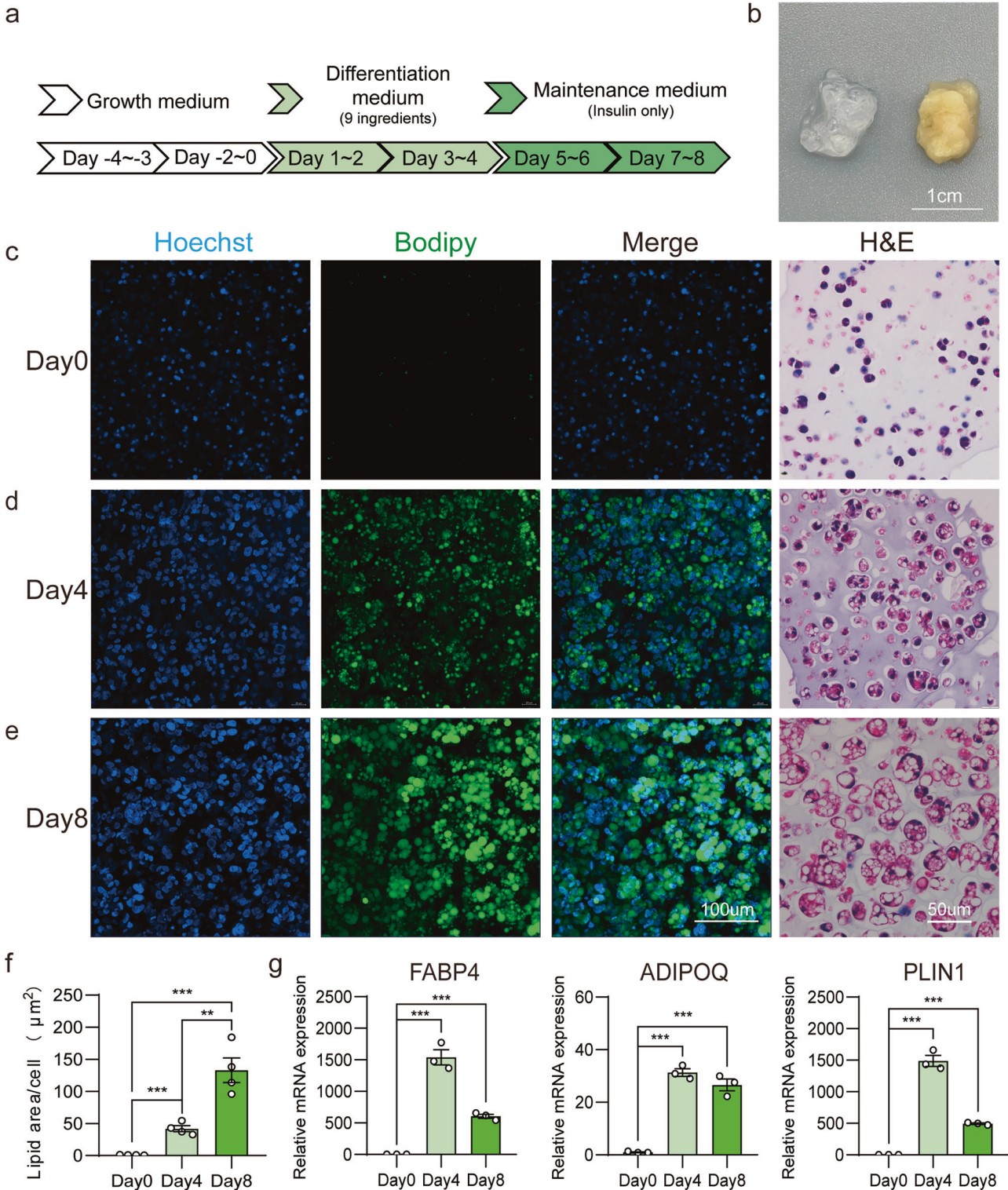

**Fig. 3 Adipogenic differentiation of ISP-4 in alginic hydrogels. a** Workflow overview: ISP-4 was embedded in alginic hydrogels, cultured in the growth medium for 4 days, and subsequently differentiated using a standard 4 + 4 protocol for adipogenesis. **b** Photographs of empty alginic hydrogel (left) and ISP-4 cultured fat (right) after differentiation, scale = 1 cm. **c–e** Representative images of ISP-4 cultured fat at indicated days. Fluorescence images were taken by maximum intensity projection with confocal microscopy, green = BODIPY, blue = Hoechst, scale bar = 100 μm. Images of H&E staining were taken by a light microscope, scale bar = 50 μm. **f** The lipid content of cultured fat was quantified by measuring the BODIPY and Hoechst fluorescence area in figures (**c–e**). Data were collected by ImageJ (*n* = 4 biologically independent samples). **g** Expression levels of adipose-specific genes in ISP-4 cultured fat at indicated days. Data were measured with RT-qPCR and normalized against Day0 (*n* = 3 biologically independent samples). All numerical values are presented as mean ± SEM. Two-tailed unpaired Student's *t*-test *p*-values are indicated as **$p \leq 0.01$, ***$p \leq 0.001$, ****$p \leq 0.0001$.

significant amount of small lipid droplets appeared in the hydrogel (Fig. 3d), and lipid droplets dramatically accumulated at the end of the 4 + 4 differentiation (Fig. 3e). Quantification of the lipid droplet area revealed a 100-fold increase in lipid accumulation along the differentiation process (Fig. 3f), and the expression of adipose-specific genes was highly upregulated when the ISP-4 cultured fat was matured (Fig. 3g). Additionally, ISP-4 could still differentiate in alginate hydrogel even as the passages number increased (Supplementary Fig. 5), and other porcine preadipocyte cell stains established in this study could also be differentiated in alginate gel with 4 + 4 protocol (Supplementary Fig. 6).

Given that ISP-4 could be differentiated with the AMM only (2 × 5 protocol) or with 2% FBS ADM/AMM in a 2D culture (Fig. 2), we hypothesized that ISP-4 could also mature in 3D hydrogels using these safer and cheaper methods (Fig. 4a). After treating the ISP-4 alginate fibers with the 2 × 5 protocol, the cultured fat accumulated significantly less lipid than those treated with the standard 4 + 4 protocol (Fig. 4b, c). Moreover, the expression of adipose-specific genes in the 2 × 5 protocol was also less than 4 + 4 protocol (Fig. 4d). Interestingly, the cells formed clumps in the cultured fat differentiated with the 2 × 5 protocol (Fig. 4b), implying that the cells had not completely lost their proliferation ability and continued to grow during differentiation in AMM. In contrast, reducing FBS to 2% within the 4 + 4 protocol did not change the cell distribution in the alginate hydrogel but significantly reduced lipid accumulation (Fig. 4b, c). As expected, the expression of adipose-specific genes was also reduced in the cultured fat differentiated with 2% serum (Fig. 4d).

Although alginate is a commonly used natural polymer in the food industry and cell-cultured fat, it cannot provide the protein and fibrous texture of real meat. As an alternative, textured soy protein (TSP) is an inexpensive and porous protein byproduct from oil production that has been successfully used as a bioscaffold in beef muscle cell-cultured meat[17]. So, we asked if ISP-4 could grow and differentiate in TSP. TSP slice was seeded with $2 \times 10^6$ ISP-4 cells, and adipogenesis was stimulated using the standard 4 + 4 protocol. The differentiated TSP slices showed a yellow hue similar to alginate cultured fat (Supplementary Fig. 7a), and they also floated on 4% PFA solution, possibly due to decreased density resulting from accumulated lipids (Supplementary Fig. 7b). However, analysis of H&E stained sections revealed that only a few ISP-4 cells colonized inside the TSP slice, with most cells forming a layer outside the TSP slice and accumulating relatively less lipid than in alginate hydrogel (Supplementary Fig. 7c). These results indicate that further optimization of the TSP-based cultured fat protocol is necessary.

**ISP-4 can be expanded effectively in upscaled cultures.** Parallel to the powerful adipogenic capacity, efficient expansion of seed cells is also required for a cell-cultured fat bioprocess. Upscaled cell culture can dramatically increase the efficacy of cell expansion, compared to culturing in a flask. Therefore, we analyzed the proliferative capacity of ISP-4 using a microbead-based upscaled cell culture system. We employed 3D TableTrix® microcarriers in a 100 mL spinner flask culture system to expand ISP-4. After overnight incubation, ISP-4 colonized on the microbeads and rapidly proliferated during the following 6-day culture (Fig. 5a). The cell counting data indicated that the logarithmic phase occurred from day 2 to day 5, and the stationary phase appeared on Day 6, resulting in a 40-fold increase in cell density (Fig. 5b). Even at high passage numbers, ISP-4 was still able to proliferate on microcarriers at a similar rate (Fig. 5c). Furthermore, after harvesting from microcarriers, ISP-4 cells could efficiently

differentiate in alginate hydrogels, accumulating a similar amount of lipids compared to cells from 2D culture (Fig. 5d, e).

Previous studies have shown that many primary preadipocytes can grow on Cytodex 1 microcarriers[18,20]. However, the process of digesting cells from microcarriers is required since Cytodex1 is inedible, increasing workloads and costs. In contrast, the 3D TableTrix® microcarriers we used are made of porcine skin gelatin. Therefore, we attempted to in-situ differentiate ISP-4 on the bead surface. After reaching the plateau phase, we replaced the growth medium in the spinner flask with ADM for 4 days and then replaced it with AMM for another 4 days. Similar to the high adipogenesis observed in the alginate bioscaffold, lipid accumulation occurred along with the 4 + 4 adipogenic process (Fig. 6a), eventually covering the surface of microcarriers with mature adipocytes (Fig. 6b). This in-situ differentiation capability of ISP-4 was well preserved during the passage increasing (Fig. 6b–d). Collectively, these results demonstrate that the ISP-4 cell strain can be efficiently proliferated and differentiated on 3D Table-Trix® microcarriers-based upscaled culture.

**ISP-4 can co-differentiate with myoblasts to form marble-like structure.** Real meat is composed of both adipose and muscle tissues, and the marbling patterns resulting from intermuscular fat are a hallmark of high-quality meat. To mimic this marbling in cultured meat, various techniques have been developed, most of which involve separately differentiating muscle and fat cells before combining them[21–23]. However, simultaneous maturation of both cell types could significantly reduce time and cost. Therefore, we investigated whether it was possible to co-culture and differentiate ISP-4 and myoblast cells together.

To achieve this, ISP-4 was co-cultured with commonly used myoblast cells C2C12 in a growth medium for 2 days until confluence was reached. Since 2% horse serum (HS) is commonly used to stimulate C2C12 myogenesis, and ISP-4 could efficiently adipogenic differentiated with 2% FBS, the mix cells were firstly treated with 4 + 4 adipogenic protocol with 2% FBS to stimulate ISP-4 adipogenesis. After that, cells were maintained in AMM (with 2% HS) for another 4 days to induce C2C12 myogenesis (Fig. 7a).

During the first 4 days of adipogenic induction, lipid droplets were observed, and their size and number continued to increase during the subsequent 4 days of AMM treatment (Fig. 7b). Additionally, myotubes began to form on day 8 and showed accelerated growth in the 2% HS AMM medium. By the end of the treatment, brightfield microscopy revealed constructs containing mature adipocytes surrounded by muscle fibers, akin to real meat (Fig. 7b). Immunofluorescence staining further confirmed the maturation of adipocytes and myocytes (Fig. 7c). Notably, these cell types exhibited close interfacing without mutual repulsion. Fat-specific genes were upregulated after the ADM treatment and downregulated after AMM treatment, similar to the regular 2D culture (Fig. 1d). Interestingly, fat-specific genes further increased with 2% HS AMM treatment (Fig. 7d), suggesting that ISP-4 could accumulate lipids in this medium. Similarly, muscle-specific genes increased during the differentiation process, with *Myod1* and *Mhy1* peaking on day eight and then slightly decreasing, while *Myog* mRNA continued to increase throughout the 12-day differentiation process (Fig. 7e).

Since C2C12 is a mouse cell line and is unsuitable for cultured meat production, we explored the co-culturing and co-differentiation of porcine muscle satellite cells (PMSC) with ISP-4. Given that the myogenic differentiation capability of porcine satellite cells is not as robust as that of C2C12, we adjusted the differentiation protocol for ISP-4 to protect its myogenic capacity. The time in ADM and AMM was reduced to

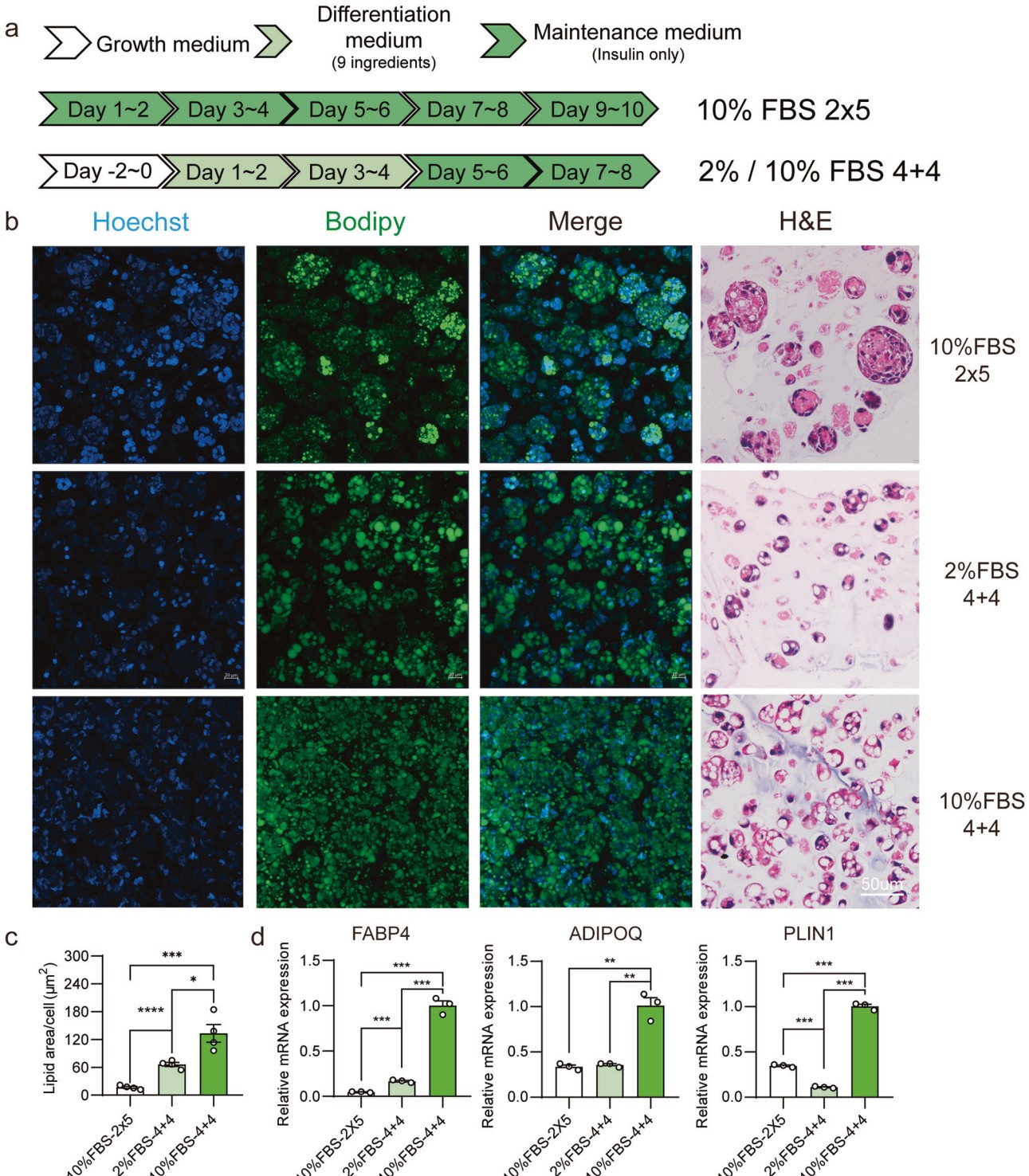

**Fig. 4 Adipogenesis differentiation of ISP-4 in alginate hydrogels using cost-effective and low-toxicity methods. a** Schematic representation of adipogenic methods for ISP-4 cultured in alginate hydrogels. b. Representative images of ISP-4 cultured fat with the indicated methods. Fluorescence images were captured using confocal microscopy with maximum intensity projection, green = BODIPY, blue = Hoechst, and scale bar = 100 μm. Light microscope images show H&E staining, and the scale bar = 50 μm. **c** The lipid content of the cultured fat in figure **b** was quantified by measuring the BODIPY and Hoechst fluorescence areas. Data were collected by ImageJ ($n = 4$ biologically independent samples). **d** Expression of adipose-specific genes for the samples presented in figure (**b**). RT-qPCR was used to measure the data ($n = 3$ biologically independent samples). All numerical values are expressed as mean ± SEM. Two-tailed unpaired student's t-test p-values are indicated as **$p \leq 0.01$, ***$p \leq 0.001$, ****$p \leq 0.0001$.

48 h, while 96 h in 2% HS was maintained for myogenesis (Supplementary Figure 8a). Observations showed that ISP-4 began lipid accumulation on Day 3, with the diameter of lipid droplets progressively increasing by Day 5. Muscle cells started to elongate and grow in an orderly fashion from Day 5. By Day 8, after 96 h of culture in 2% HS, the lipid droplets in the adipocytes had further expanded, and the interlaced growth of mature muscle fibers and adipocytes was evident (Supplementary Fig. 8b).

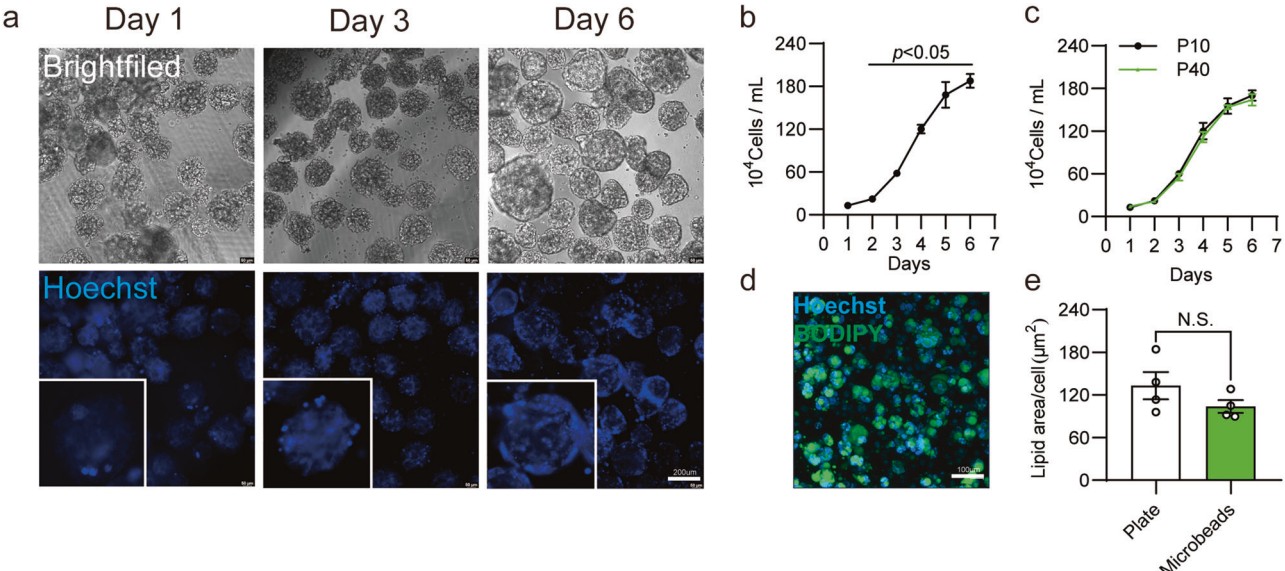

**Fig. 5 Proliferation of ISP-4 within upscale cultures. a** Representative images of ISP-4 cells on 3D TableTrix® microcarriers at indicated time points. Hoechst was used to stain the nuclei, and the enlarged images (lower left framed) show increasing spots on the microcarriers. Scale bar = 200 μm. **b** Growth curves of ISP-4 cultured on microcarriers. Cells were dissociated from microcarriers and counted with a hemocytometer (n = 3 biologically independent samples). **c** Growth curves of P10 and P40 ISP-4 cultured on microcarriers (n = 3 biologically independent samples). **d** Representative maximum intensity projection confocal image of alginic hydrogels with ISP-4 harvested from microcarriers. Adipogenesis was induced with the 4 + 4 protocol (10% FBS), and lipids were stained with BODIPY. Hoechst was used to stain the nuclei. Scale bar = 100 μm. **e** Quantification of lipid content in cultured fat with cells harvested from 2D culture or microcarriers. BODIPY and Hoechst fluorescence were measured using ImageJ from four independent samples (n = 4 biologically independent samples). All numerical values are presented as mean ± SEM. Two-tailed unpaired Student's t-test p-values are indicated.

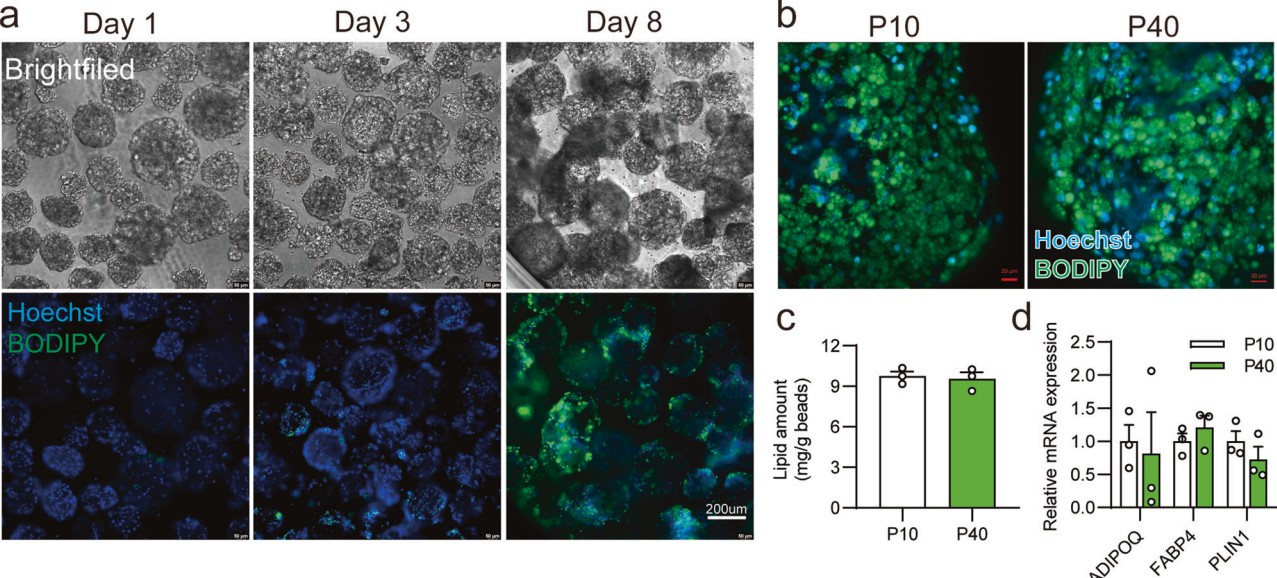

**Fig. 6 Adipogenic differentiation of ISP-4 on microcarriers within upscale cultures. a** Representative images of ISP-4 on microcarriers at indicated time points of adipogenic differentiation, scale bar = 200 μm. Lipid was stained with BODIPY, and nuclei with Hoechst, scale bar = 200 μm. **b** Representative images of adipogenic differentiated ISP-4 on microcarriers with a "4 + 4" protocol. Images were captured using confocal microscopy with maximum intensity projection. Scale bar = 20 μm. **c** Lipid contents for the samples form figure **b** (n = 3 biologically independent samples). **d** Expression of adipose-specific genes for the samples from figure (**b**). RT-qPCR was used to measure the data (n = 3 biologically independent samples). All numerical values are presented as mean ± SEM.

BODIPY and phalloidin staining further confirmed the mixed growth pattern of mature adipocytes and muscle cells (Supplementary Fig. 8c). Additionally, qPCR detection of adipose-specific and muscle-specific genes validated the differentiation and maturation of both cell types following this co-culturing and co-differentiation process (Supplementary Fig. 8d, e). Immunoblotting against DESMIN further confirmed the myogenesis of PMSC within this process (Supplementary Fig. 8f).

Collectively, we demonstrated that the remarkable adipogenic differentiation capability of ISP-4 enables it to co-culture and co-

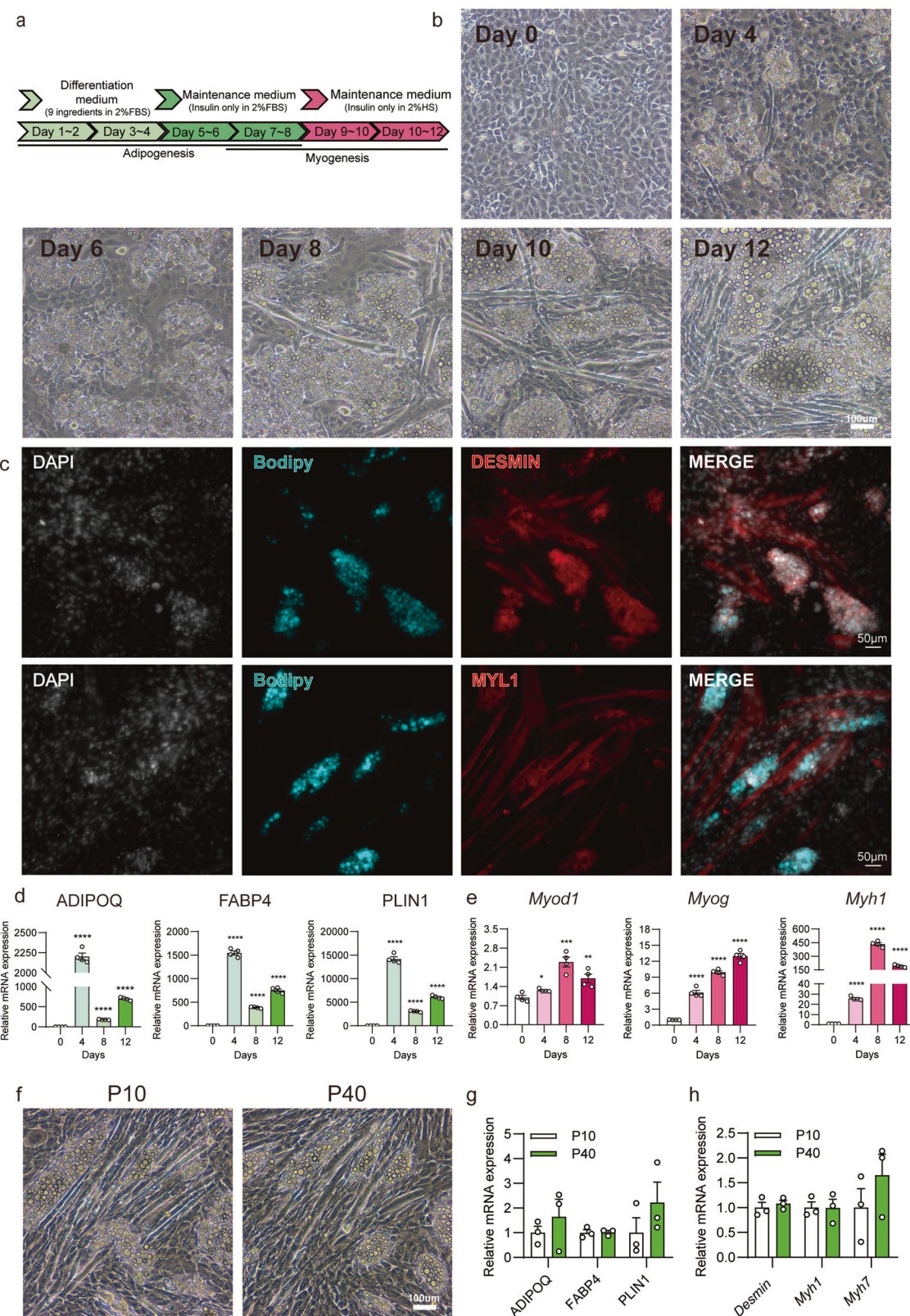

differentiate with muscle cells, thereby reducing production and labor costs.

## Discussion

The acceptance of cultured meat by consumers depends, in part, on its ability to replicate the distinctive aroma and juicy texture of traditional meat, which is primarily provided by fat[24–26]. As a result, attention has shifted towards the production of fat cells in cellular agriculture[18,27]. To produce cell-cultured fat, a steady supply of seed cells is required. Although fresh primary pre-adipocytes can always be obtained from slaughtered animals, non-inbred farm animals may produce primary cells with varying responses to the same culture condition, leading to uncertainties in large-scale production.

**Fig. 7 Co-differentiation of ISP-4 with myoblast cells. a** Workflow overview: ISP-4 was co-cultured with C2C12 cells and adipogenic differentiation was induced using a standard 4 + 4 protocol (2% FBS). Subsequently, myogenic differentiation was induced in DMEM with 2% HS for an additional 4 days. The medium was changed every 2 days. **b** Representative images of co-cultured ISP-4 and C2C12 cells at indicated time points after treatment. Notice the increasing number of lipid droplets and myotubes. Scale bar = 100 μm. **c** Representative fluorescence images of co-differentiated ISP-4 and C2C12 cells. The images were taken using confocal microscopy after immunofluorescence staining against either DESMIN or MYL1, nuclei with DAPI, and lipid droplets with BODIPY. Scale bar = 50 μm. **d** and **e** Expression of adipose-specific genes (**d**) and muscle-specific genes (**e**) for the samples presented in figure (**b**). The data were measured by RT-qPCR and normalized against day 0 (n = 4 biologically independent samples). **f** Representative images of co-cultured P10 and P40 ISP-4 and C2C12 cells after differentiation. Scale bar = 100 μm. **g** and **h** Expression of adipose-specific genes (**g**) and muscle-specific genes (**h**) for the samples presented in figure (**f**). The data were measured by RT-qPCR (n = 3 biologically independent samples). All numerical values are expressed as mean ± SEM. Two-tailed unpaired Student's t-test p-values are indicated as **$p \leq 0.01$, ***$p \leq 0.001$, ****$p \leq 0.0001$.

In this study, we presented an immortalized porcine cell strain that could subculture over 40 passages without significant attenuation of its adipogenic potential. This cell strain might be the first immortalized porcine adipogenic cell strain that could be used in a cell-cultured fat study.

Many methods have been developed for cell immortalization, which are generally divided into three categories: overexpression, gene knockout, and spontaneous immortalization[28]. Over-expression represents the most prevalent and mature method for achieving cellular immortality, usually expressing telomerase (TERT), SV40T, or HPV-E6/E7. It is undeniable that cultured fat with exogenous protein expression can impact consumer choices regarding cultured meat due to food safety concerns[29-31]. However, genetic modification can also provide numerous benefits such as higher differentiation potential[32,33], lower production costs[34], and even strengthened nutrition[35].

In the wake of CRISPR/Cas9 technology, the prospect of knocking out tumor suppressor genes (such as p53, Rb, and others) to attain cellular immortalization has become a tangible reality. However, to this date, there is no report of seed cells being immortalized through gene knockout. But this solution can indeed eliminate the food safety risks caused by overexpression. Nonetheless, it is critical to note that gene knockout, in a broad context, is still categorized as genetically modified food and may thus remain subject to regulations[36].

Spontaneous immortalization is relatively the safest immortalization method. Recent research by Pasitka et al. showed that immortalized chicken fibroblast cell lines could be transdifferentiated into adipogenic seed cells by PPARγ-specific agonists, indicating that spontaneous immortalized cells are excellent seed cells for cultured meat[19]. However, it should be noted that spontaneously immortalized cells may not uniformly possess the capability to differentiate into adipocytes or muscle cells. Therefore, it's crucial to incorporate the methodologies used in this research to select cell strains with enhanced differentiation potential.

To differentiate the porcine cell line with limited adipogenesis potential, two media recipes are typically used in the two-step protocol to achieve adipogenesis: adipogenic induction medium (AIM) and adipogenic maintenance medium (AMM). Unfortunately, the chemicals in AMM, including IBMX, dexamethasone, and rosiglitazone, are regulated pharmaceuticals and can be unsafe for food if overdosed. To differentiate the porcine cell line with limited adipogenesis potential, researchers usually maintain the cells in ADM until lipid droplets appear, which causes a significant food safety risk[11,27]. However, our data demonstrated that ISP-4 could accumulate a considerable amount of lipids in only AMM, eliminating the need for hazardous drugs and achieving a substantial food safety advantage for future applications.

Reducing the cost of serum for cell culture is crucial for the price superiority of cell-cultured meat. Currently, over 95% of researchers use 10-20% FBS during preadipose cell line expansion and differentiation, which is expensive[15]. ISP-4 has been shown to accomplish adipogenesis within 2% FBS in Petri dishes and on bioscaffold. Although 2% FBS did not reduce the lipid amount in the 2D culture (Fig. 2), it significantly impaired lipogenesis in alginate microfiber (Fig. 4), indicating that the energy preference of ISP-4 for de novo lipogenesis varied under different culture conditions. Future work should focus on optimizing the medium recipe to reduce serum usage or even using the serum-free medium for ISP-4 expansion and differentiation.

For large-scale expansion of seed cells for cell-cultured meat, acclimating to 3D-scale culture is critical. Several primary pre-adipocyte cell lines have been reported to be amplified on microcarriers in suspension bioreactors to increase scalability[18,20]. We reported that ISP-4 could achieve large-scale expansion on microcarriers at the density of $18 \times 10^5$ cells per ml. However, the cell density has not yet reached the threshold for the commercial viability of cultured meat, which is $50 \times 10^6$ cells per ml[37]. Therefore, further work should focus on adapting ISP-4 to anchorage-independent growth.

Moreover, the primary preadipocytes need to be harvested from microcarriers before adipogenic differentiation[18,20]. Although it remains unclear whether it is because the microcarriers are inedible, or these primary cells cannot be differentiated on the microcarriers, the inadequacy of these protocols is apparent, which increases the labor costs. In contrast, ISP-4 could be differentiated on edible microbeads, which could significantly improve the cost advantage of cell-cultured fat. However, in this study, we used microcarriers made from pork skin, which contradicts the goal of reducing reliance on animal components. Therefore, other edible microcarriers should be tested if they go with ISP-4 in the future.

In natural meat, fat cells exist in the form of intermuscular fat along with muscle cells to form marbling. However, in vitro co-culture experiments have revealed that muscle and fat cells inhibit each other's differentiation process[38-42]. Consequently, scientists have resorted to differentiating myoblasts and preadipocytes separately and then assembling them into structured meat using 3D printing or direct stacking techniques[21-23]. However, these methods are time-consuming and labor-intensive, which hampers their efficiency. Therefore, to enhance efficiency, it is crucial to obtain a preadipocyte cell strain that can co-differentiate with myoblast cells. In this study, we investigated the adipogenesis potential of ISP-4 in 2%FBS and successfully co-differentiated adipose and muscle cells together. However, since C2C12 is a rodent cell, further research is required to examine if ISP-4 can co-differentiate with porcine myoblast cells.

Overall, in this study, we characterized an immortal porcine preadipocyte cell strain, ISP-4. It provides a potentially high-quality seed cell for cell-cultured meat production and offers a valuable model for cell-cultured meat research. Furthermore, we demonstrated that single-clone pickup from immortal cells exhibits stronger differentiation capabilities compared to using mixed primary cells.

However, this research has its limitations. One of the limitations is the immortalization of cells using lentiviral expressing SV40T, which could raise potential food safety concerns. Another limitation is the continued usage of animal-derived products, which we have not managed to completely eliminate. Finally, due to equipment constraints, the production performance of ISP-4 was not validated in a larger-scale system.

Future studies should prioritize the creation of spontaneously immortalized preadipocyte cell strains across a broader range of species. Additionally, selecting more single-cell strains to establish seed cell banks with diverse characteristics would be beneficial. By optimizing the culture medium formula, the use of animal-derived products can be minimized and production costs can be further reduced.

## Methods

**Cell culture**. ISP-4, C2C12, and immortalized porcine muscle satellite cell (PMSC, a kindly gift from Professor Li-Min Hou, Nanjing Agriculture University) were cultured in growth medium (GM), containing Dulbecco's modified Eagle's medium (Servicebio#G4511), 10% fetal bovine serum (Procell#164210-50), 100U/ml penicillin (Sangon#A610028), 100 μg/ml streptomycin (Sangon#A610494). For PMSC, an extra 10% FBS and 2 ng/mL FGFb (Novoprotein#C046) were added.

ISP-4 adipogenic differentiation medium (ADM) consisted of growth medium supplemented with 1 μg/mL Insulin (Novolin R), 5 nM 3,3',5-Triiodo-L-thyronine (Yuanye Bio-Technology#S24025-25mg), 2 μg/ml dexamethasone (Sangon#A601187), 100 μM 3-isobutyl-1-methylxanthine (IBMX, Sangon#A606630), 125 μM indomethacin (Makclin#I811784), 1 μM rosiglitazone (Makclin#R832516), 33 μM biotin (Makclin#B6220), 17 μM Pantothenic acid (Sangon#A600683), 1 μg/μl Transferrin (Yuanye Bio-Technology #S12027). ISP-4 adipogenic maintenance medium (AMM) includes growth medium supplemented with 1 μg/mL Insulin (Novolin R).

For the 4 + 4 protocol, the cells were first recovered in a growth medium for 2 days, followed by a switch to ADM for 4 days. Afterward, the cells were maintained in AMM for an additional four days, mediums were changed every 2 days.

For the 2 × 5 protocol, the cells were seeded and cultured in AMM for 10 days, with medium changes every 2 days.

**Cell culture in alginate hydrogel**. To prepare the cell/alginate suspension, ISP-4 cells were resuspended in a 0.75% alginate solution (Sigma#W201502) at a concentration of $1.25 \times 10^7$ cells/ml. The suspension was then injected into a solidification buffer (50 mM $CaCl_2$) and placed in a 37 °C incubator for 20 minutes to solidify into microfibers. After washed with DMEM, the microfibers were transferred into a growth medium (GM) for recovery before adipogenesis according to different protocols.

**Cell culture in wire-drawing protein scaffold**. Irradiated sterilized wire-drawing peanut protein was soaked in growth medium overnight and then cut into small cubes (about 1 cm² in area and 0.3 cm in thickness). These cubes were subsequently dried using sterile filter paper and placed onto a 10 cm dish for later use.

To seed cells onto the scaffolds, $2.5 \times 10^6$ ISP-4 were resuspended in 7 μl thrombin (20NIH units per ml, Biosharp#BS903). Next, 7 μl fibrinogen (15 mg/ml, Biosharp#BS943) was added immediately before seeding the cells onto the scaffolds. The cell blend was mixed well and dropped onto the scaffolds. The dish with seeded scaffolds was then placed into a 37 °C incubator for 20 minutes to allow the fibrin to gel. Once gelation had occurred, a growth medium was added to the dish, and the scaffold with cells was cultured as indicated.

**RNA extraction, cDNA synthesis, and RT-qPCR**. RNA isolation was performed using Total RNA Extraction Reagent (Vazyme#R401), followed by reverse transcription using HiScript II Q RT SuperMix (Vazyme#R223), as per the manufacturer's guidelines. Real-time quantitative PCR (RT-qPCR) was carried out with iQ SYBR Green Supermix (Servicebio#G3326) using the QuantStudio™ 5 System (ThermoFisherScientific). The sequence of primers is listed in supplementary Table 1. The $2^{-\Delta\Delta ct}$ method was used to calculate relative gene expression, with 36B4 as the reference gene for normalization.

**Fluorescence staining**. For 2D cultured ISP-4 or mixed culture cells of ISP-4 and C2C12/PMSC, samples were fixed in 4% paraformaldehyde. After washing with PBS, fixed cells were permeabilized with 0.2% Triton X-100, then blocked with 3% BSA solution, anti-DESMIN (Abclonal, Cat#A0699, Lot#5500004718) and anti-MYL1 (Servicebio, Cat#GB112474, Lot#AC230928017) were used to detect mature muscle cells. After being washed with PBS, cells were re-stained within Goat Anti-Rabbit IgG-DyLight 549 (Bioworld, Cat#BS10023, Lot#CL89330), Hoechst33342 (1:1000, Beyotime#C1028), and BODIPY FL (1:5000, Thermo Fisher#D3922).

For alginate hydrogel, samples were fixed in 4% paraformaldehyde containing 50 mM $CaCl_2$. The samples were stained with Hoechst33342 (1:1000, Beyotime#C1028), BODIPY FL (1:5000, Thermo Fisher#D3922) and Actin-Tracker Red-Rhodamine (1:200, Beyotime#C2207S) in PBS for 30 min. And the samples were washed with PBS before imaging.

**ISP-4 and C2C12/PMSC mixed culture**. For coculture with C2C12, a suspension of ISP-4 cells ($1 \times 10^5$ cells) and C2C12 cells ($3 \times 10^5$ cells) was seeded in six-well plates containing growth medium. The cells were then cultured for 48 h. Subsequently, differentiation was initiated by switching to ADM, and after 4 days, the medium was exchanged into AMM for another 4 days. Then the medium was changed into AMM with 2% horse serum to initiate myogenesis.

For coculture with PMSC, $0.6 \times 10^5$ ISP-4 cells and $1.4 \times 10^5$ PMSC cells were seeded into a 12-well plate with 1 mL culture medium. After 24 h recovery, cells were differentiated with ADM for 48 h, and AMM for another 48 h. Then cells were cultured in AMM with 2% horse serum for the other 4 days.

The cells were maintained in a 5% $CO_2$ humidified incubator at 37 °C, with medium exchanges every 2 days.

**Microcarrier cell culture**. 100 mg of 3D TableTrix microcarriers (CytoNiche) were dissolved in 20 mL of growth medium in a 125 mL sterile spinner flask (CytoNiche) to obtain a final concentration of 5 mg/mL. The mixture was left to dissolve overnight at 4 °C. The next day, the spinner flasks were prewarmed to 37 °C, and a total of $2.5 \times 10^6$ ISP-4 cells in 30 ml growth medium were added.

The spinner flasks were positioned on magnetic stirring platforms in the incubator. For the first 24 h, rotation speed was set to 35 rpm for 5 min, then followed by 0 rpm for 1 h, and repeated 24 times. Afterward, the rotation mode was changed to a constant speed of 40 rpm. 50% growth medium was changed every 48 h.

For cell counting, 1 mL of medium containing microcarriers was collected from a spinner flask. 200 μL of supernatant was removed and replaced with lysate (CytoNiche). After incubating at 37 °C for 30 min, the cells were then counted using a hemocytometer,

To assess the differentiated microcarriers in culture flasks, 1 mL of cell suspension was pipetted and stained with

Hoechst33342 (1:1000, Beyotime, C1028) and BODIPY FL (1:5000, Thermo Fisher, D3922).

**Area of lipid droplet analysis**. All images were taken by confocal laser scanning microscope (LSM750, Zeiss) under the same acquisition conditions. Images were processed in ImageJ (version: 2.1.0/1.53c) as the following step. 1. Scale bar calibration: Open the image with only BODIPY staining, measure the length of the image's scale bar, and set the pixel-to-length (μm) ratio in "set scale". 2. Fluorescence area measurement: Convert the image to "8-bit", open "Threshold", adjust the threshold, select the green fluorescent area, and select "Measure" to obtain the area of the lipid droplets. 3. Cell nuclei counting: open the image with only nucleus staining, and use ImageJ's "analyze particles" tool to count the cell nuclei stained with Hoechst.

**Oil-red staining**. Cells were fixed with 4% PFA at 4 °C overnight. Following fixation, the cells were rinsed with 60% isopropanol and then stained with Oil Red O (0.42 g/mL in 60% isopropanol) for 5 min. The Oil Red O solution was subsequently removed, and the cells were washed three times with tap water. The Oil Red O stain was then extracted using isopropanol and the absorbance was measured at 500 nm.

**Western blot**. Protein was extracted with RIPA (Servicebio#G2002), target proteins were detected with antibodies as follow: anti-PERILIPIN 1 (Bioss, Cat#bs-3789R, Lot#AD121123), anti-DESMIN (Abclonal, Cat#A0699, Lot#5500004718), Anti-FLAG (Genscript, Cat#A00187, Lot#155000952), Anti-GAPDH (Bioworld, Cat#AP0063, Lot#AA55151), anti-TUBULIN (Sangon, Cat#D190090-0100, Lot#H508AA0104).

**Lipid content analysis**. The lipid content was measured using a triglyceride content assay kit (Sangon#D799795), based on colorimetric method, following the manufacturer's instructions. For sample preparation, ~0.1 g of hydrogel or microcarrier was homogenized using a blue pestle in 1 ml Buffer 1. Then, 200 μL of supernatant was collected for further analysis. 200 μL solution with 1 mg/mL triglyceride was used for comparison. The lipid content was calculated using the following formula.

$$\text{Lipid content} = (1\,\text{mg/mL} \times 0.2\,\text{mL})$$
$$\times \frac{A^{\text{sample}} - A^{\text{Blank}}}{A^{\text{stander}} - A^{\text{Blank}}} / \text{sample weight (g)}$$

**Statistics and reproducibility**. Statistical analysis was performed using Prism 9.0 (GraphPad). Analysis between two groups was performed using a Two-tailed unpaired Student's *t*-test. Sample size has been indicated in each figure legend.

**Reporting summary**. Further information on research design is available in the Nature Portfolio Reporting Summary linked to this article.

## Data availability

The source data can be found in Supplementary Data 1. The uncropped immunoblottings can be found in Supplementary Figs. 1 and 9. All other data that support the findings of this study are available from the corresponding authors depending on reasonable request.

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

## Acknowledgements

We thank Ms. Jiping Wu from Cytoniche Biotechnology for kindly providing training for upscale culture equipment, and Dr. Li-Min Hou for kindly providing immortal porcine muscle satellite cells. This work was supported by the National Natural Science Foundation of China No. 32170847, and the Fundamental Research Funds for Central University No. YDZX2023004 to Z.H.

## Author contributions

Experiment performing: Y.-M.C., P.-C.H., M.-M.S., H.-N.Z., J.Q., Z.-D.Z., Z.H. Data analysis: Y.-M.C., H.C., X.-Z.M., M.-Y.T., Z.H. Manuscript writing: Y.-M.C., Z.-D.Z., Z.H. Project design: W.-Y.Z., Z.H.

## Competing interests

The authors declare no competing interests.
