## [Peer review file · Communications Biology]

Reviewers' comments:

Reviewer #1 (Remarks to the Author):

This study reports on the usefulness of the porcine adipocytes to produce cell-cultured fat in cultured meat. As the authors claims, the proliferation and differentiation ability of the adipocyte is directly related with the efficiency of cultured meat productions, and this study demonstrated that this immortalized adipogenic cells maintained these two important abilities. However, the optimization of the culture method was insufficient. Although the title of the result section is "optimizing methods for Pig4#", it was not comprehensive in this study. In addition, the application parts (gel encapsulation, microcarrier, and co-culture) were superficial. As the authors cited, previous studies have already reported similar results to the parts in this paper. Therefore, this study needs to focus on the advantages of the immortalized adipocytes, compared with the other adipocytes used in the previous studies. For example, when the results in Figure 5 and 6 using the cells at P10 was compared with those using the cells at P40, they showed some difference? All the data shown in this paper can be obtained from the cells at P10 and P40? Since the gel formation technique, the use of microcarriers, co-culture with muscle cells have been reported in previous studies, the authors should clarify the uniqueness of this study in the application parts.

In this study, adipogenic capability was analyzed by imaging using BODIPY. The authors should explain in this manuscript why the capability was able to be analyzed by the method. In addition, to analyze it quantitatively, another experiments are required to directly detect the expression amount of adipogenic proteins, since the adipogenic capability of the cells at P10 and P40 is one of the most important factor in this study.

The authors need to explain how to decide the condition of stirring culture with microcarriers (e.g., rotation speed, cell seeding density). The condition was optimized specifically for the cells produced in this study? Is it the same as the method reported previously (references 18, 20)? Which part was developed by the authors? Again, the cells at P10 and P40 show the similar behavior in this experiment?

As the authors described in the manuscript, C2C12 is mouse muscle cell line. In this study, on the other hand, adipocytes were produced from an appropriate animal source for cultured meat production, and the authors claimed the advantage of this study. C2C12 have been widely used in a large number of previous studies, since it is easy to induce myotube differentiation. In addition, the mouse muscle cells and porcine fat cells were just co-cultured. Although the both cells showed differentiation into myotubes and adipocytes in the culture dish, they didn't show the marbling pattern. This co-cultured cell assembly was not similar to real meat consisting of myofibers and fat components.

Unpublished data should be not used to discuss the study since the readers cannot judge the reliability of the statement.

Reviewer #2 (Remarks to the Author):

Overall:

This study is certainly in an important area, and the science presented is generally of good quality, for which I would commend the authors.

However, whilst it certainly is an interesting observation that an immortalised porcine cell line retained adipogenic differentiation potential for many generations, this is not an especially profound finding, and the additional characterisation of this observation presented here, is (in this reviewer's opinion) not sufficient to merit publication in a journal such as Communications Biology.

Several significant further elements would need to be added, in this reviewer's opinion, to demonstrate the potential of this cell line for use in cultured meat research and production

bioprocesses.

Major Comments:

The use of many animal derived products, principally FBS in the medium (for both proliferation and differentiation), and porcine gelatin-derived microcarriers, are incompatible with accepted philosophies for cultured meat production. Whilst the presented results around upscaled cultures are promising, they do not yet represent an actual proof-of-principle for this cell line. Various defined media formulations have been developed (and are in the public domain), at the very least a selection of these should be tested in short-term proliferation assays.

Whilst the acceptability of genetic modifications of different types is rather unclear for cultured meat, and will certainly vary between jurisdictions, the method of immortalisation presented here (use of an SV40 large T antigen) is amongst the most contentious, given that it uses viral transgenic material. The authors do allude to other possibilities for immortalisation in their discussion, but in my opinion, some actual data is needed here. It seems likely that a combination of telomerase and CDK4, or telomerase together with knockout of p16, might be sufficient for immortalisation.

The use of rodent C2C12 cells, which are known to differentiate extremely robustly, for the co-differentiation experiments, is unfortunately not a particularly impressive proof-of-principle when it comes to co-culture. Porcine primary cells (even if early passage, not immortalised) would have been much more informative.

The adipogenic differentiation media contains several substances of concern (as noted by the authors in their discussion). Porcine FAP differentiation has recently been demonstrated with a simplified formulation (Mitic et al., 2023), which the authors should consider testing.

Figure 2: The 2x5 and 4+4 medium strategies are hard to compare as they have different amounts of time in growth and differentiation medium. Better controls would have been good to see, but I suppose the authors are comparing the medium strategy and not doing a direct comparison of the two differentiation media. I would have also liked to see a negative control without inducers, which would have helped to show if 2x5 is mostly just causing 'passive' lipid accumulation or true adipogenesis.

Minor Comments:

Line 223. The resulting Pig#4 cultured fat dramatically differed from the empty alginate hydrogel in appearance, the yellow hue of Pig#4 cultured fat is similar to other cultured fat with primary cells reported before (Fig. 3B).

It would be interesting here to see a comparison with undifferentiated cells.

Line 230. Interestingly, most cavities in the alginate were occupied by 2-3 cells, indicating that Pig#4 may divided in the alginate hydrogels (Fig. 3C).

This seems rather speculative, the authors might consider EdU staining or try DNA quantification to explore this further.

Figure 4: The 10% FBS 4+4 confocal image is horribly overexposed and needs to be retaken.

The statistical analysis and data presentation is generally sound, and the descriptions of the methodology are sufficiently detailed.

Reviewer #3 (Remarks to the Author):

Title: An immortal porcine preadipose cell strain for efficient production of cell-cultured fat
The focus of the manuscript is the development of immortalized pre-adipose cells and the optimal culture conditions for the production of cultured meat. The manuscript discusses several aspects related to pre-adipose cell line Pig#4, including its potential for differentiation, the optimal adipogenic method, its proliferation and differentiation in 3D culture, and its co-differentiation with rodent myoblasts (C2C12). This manuscript is commendably well organized and written, with a conclusion that appropriately derives from the experimental data. However, it is important to acknowledge the limitations of this study, such as the use of immortalized cells and rodent cells for co-differentiation. Despite these limitations, the study's overall impact on cell culture technique and the utilization of edible microcarriers to reduce the cost of FBS is undeniably impressive. Nonetheless, there are a few concerns and suggestions that could enhance the manuscript further. However, there are some concerns and suggestions for improving the manuscript.

Major concerns

- Please consider including a discussion on the limitations of this study and proposing future directions for more meaningful, edible, and cost-effective cell-cultured fat. This would provide valuable insights for further research.
- The 2x5 protocol is an impressive method as it avoids the use of potentially toxic differentiating agents. However, the mRNA expressions of the 2x5 protocol are higher than those of the 4+4 protocol in Fig. 2D. In addition, in Fig. 2C, the lipid accumulation of the 4+4 protocol is higher than that of the 2x5 protocol, but the mRNA expressions of adipose-specific genes (FABP4, PLIN1, and AdipoQ) are lower. Please explain the difference between the fat accumulation data and the gene expression data.
- In Fig. 6D, the mRNA expressions of adipose-specific genes (FABP4, PLIN1, and AdipoQ) increased approximately two-fold on day 12 compared to day 8. The expressions of adipose-specific genes decreased when cells were treated with adipogenic maintenance medium (AMM) between days 4 and 8, but increased between days 8 and 12 when AMM (with 2% horse serum) was treated. The authors should provide a more comprehensive explanation regarding the increase of adipose-specific genes on day 12.
- To utilize in cultured meat production, it is crucial that the cells adhere well to the 3D structure. However, in the textured soy protein (TSP) used in the manuscript, the cell adhesion seems to be inadequate. Is there an experiment to evaluate the adhesion ability of Pig#4 and other cells? Alternatively, have the authors explored any methods to enhance the adhesion ability?

Minor concerns

- The authors should standardize the gene abbreviations by choosing either uppercase or lowercase letters and using them consistently throughout the figures.
- Keywords contain a mix of capital and lowercase letters.
- L67: Wewe

Responses to Reviewers' comments

Reviewer #1: This study reports on the usefulness of the porcine adipocytes to produce cell-cultured fat in cultured meat. As the authors claims, the proliferation and differentiation ability of the adipocyte is directly related with the efficiency of cultured meat productions, and this study demonstrated that this immortalized adipogenic cells maintained these two important abilities. However, the optimization of the culture method was insufficient. Although the title of the result section is "optimizing methods for Pig4#", it was not comprehensive in this study. In addition, the application parts (gel encapsulation, microcarrier, and co-culture) were superficial. As the authors cited, previous studies have already reported similar results to the parts in this paper. Therefore, this study needs to focus on the advantages of the immortalized adipocytes, compared with the other adipocytes used in the previous studies.

Response: Thank you very much for your constructive comments on our manuscript. We appreciate your insights and agree with your suggestions.

As you've rightly pointed out, there is indeed potential to further optimize the cultivation and differentiation methods for Pig#4 in the context of cell-cultured fat production. However, considering the current food regulations and public acceptance, the immediate application of immortalized cell lines for food production may not be feasible. Therefore, the primary objective behind the establishment of Pig#4 is to provide a convenient model for scientific research into cell-cultured fat. This enables scientists to further refine various methods based on their specific research goals.

Indeed, many previous studies have reported the methods we used in this research. However, our main goal was not to establish new methods for producing cell-cultured meat, but rather to validate a method for establishing seed cells with highly adipogenic potential, namely, by immortalizing and then selecting clonal cell lines. The methods we used in this manuscript were to demonstrate Pig#4 do have this potential for being seed cells.

In fact, we have already received several inquiries about Pig#4 from researchers who have read our preprint paper.

To compared with the other adipocytes used in the previous studies, we found a limited number of immortalized swine preadipose cell lines have been reported (1, 2). Regrettably, we could get neither of them, which led to the establishment of our cell strain, Pig#4. This situation precluded a direct comparison with the previously reported cell lines. However, when we juxtapose our findings with data from the studies above, it is evident that our cell line demonstrates clear advantages in both induction efficiency and induction time.

Respond Fig.1 Pig#4 is superior in adipogenic efficiency and time to published porcine adipogenic cell lines.

- A. Oli-Red stained porcine dedifferentiated fat cell lines after 12 days differentiation, data from Peng *et al.*, 2015., Scale bar=50µm.
- B. Representative image of P35 Pig#4 cells after the “4+4” protocol differentiation for 8 days. Scale bar=25µm
- C. Whole mount view of Oil-Red stained DFAT-P cells after 16 days differentiation, data from Nobusue *et al.*, 2010. Noticing the uneven distribution of red staining.
- D. Whole mount view of Oil-Red stained Pig#4 after the “4+4” protocol differentiation for 8 days.

1. For example, when the results in Figure 5 and 6 using the cells at P10 was compared with those using the cells at P40, they showed some difference?

Thank you for your insightful comment regarding the potential differences between juvenile and aged cells. We agree that this is a crucial point that needs to be addressed in our study.

In response to your query, we conducted additional experiments using Pig#4 cells at P10 and P40. Our results indicated that there were no significant differences in the growth performance of the cells on microcarriers between these passages (Fig.5C). Similarly, their ability to directly differentiate on microcarriers was also comparable (Fig.6B-D). These new findings reinforce our initial results and further validate our method.

2. All the data shown in this paper can be obtained from the cells at P10 and P40?

We appreciate your insightful commentary and share your emphasis on the importance of verifying the continued adipogenic potential of Pig#4 for the production of cell-cultured fat, even at elevated passage numbers.

As elucidated in our previous response, we have provided evidences that both high-passage and low-passage Pig#4 cells can be effectively proliferated and differentiated on edible microcarriers. Moreover, we have affirmed that high-passage Pig#4 cells show comparable results in other experiments we performed.

- a) For instance, we utilized Oil Red O staining and Western blotting to substantiate that even at P40, Pig#4 cells still retain their robust adipogenic capabilities (Fig. 1E-F).
- b) In Supplementary Figure 4, we have demonstrated that P40 Pig#4 cells exhibit adipogenic differentiation potential within alginate scaffolds that is comparable to that of their P10 counterparts.
- c) In Figure 7 F-H, we further confirmed that P40 Pig#4 cells retain their ability to differentiate into myoblasts.

In summary, our research affirms that the adipogenic potential of Pig#4, a critical ability for cell-cultured adipose tissue production, does not diminish with increasing passage numbers.

3. Since the gel formation technique, the use of microcarriers, co-culture with muscle cells have been reported in previous studies, the authors should clarify the uniqueness of this study in the application parts.

Thank you for your valuable comment. While each individual element of our methodology may have been previously reported, the unique contribution of our study lies in the innovative combination and application of these techniques: the ability to proliferate and differentiate in alginate gel, on edible microcarriers, and co-culture with muscle cells. This combination of attributes has not been reported in any other cell line in the existing literature.

Here are the two key aspects that underscore the uniqueness of our work:

- a) Our work pioneers the demonstration that the highly adipogenic seed cells can grow and differentiate on edible microcarriers. This is a significant breakthrough, potentially reducing labor costs in the production of cell-cultured meat, a feature not showcased in previous studies.
- b) Our study uniquely develops a co-culture system facilitating the growth and sequential differentiation of both adipose and muscle cells. This achievement stands out as previous studies differentiated muscle and adipose cells separately before combining them to form cultured meat. Our approach, in contrast, allows for a more integrated and efficient method of cultured meat production.

We believe these unique aspects of our research not only set our work apart, but also contribute meaningful advances to the field.

4. In this study, adipogenic capability was analyzed by imaging using BODIPY. The authors should explain in this manuscript why the capability was able to be analyzed by the method. In addition, to analyze it quantitatively, another experiments are required to directly detect the expression amount of adipogenic proteins, since the adipogenic capability of the cells at P10 and P40 is one of the most important factor in this study.

Thank you for your insightful comments. Indeed, in recent years, imaging and quantifying the fluorescent area after BODIPY staining has been a common method employed to represent lipid accumulation, which indirectly reflects the adipogenic capability of cells (3-5).

However, we agree that this method provides indirect evidence and its quantitative analysis might not be sufficient to fully support our conclusion. Therefore, to enhance the robustness of our findings, we have incorporated additional experiments. In Figure 1, we used Oil Red O staining, which is another reliable method for visualizing lipid accumulation during 2D differentiation. Moreover, we have supplemented this visual evidence with quantitative analysis by assessing the expression of PLIN1, a well-known marker for lipid droplets, using immunoblotting.

In addition to these, in Fig.6 C and SupFig. 4D, we have quantitatively evaluated the adipogenic level by measuring the triglyceride content, which is a direct biochemical marker of adipogenesis.

We believe these additional experiments, providing both qualitative and quantitative evidence, strengthen our conclusion that the adipogenic capability of cells at P10 and P40 have not changed. We appreciate your suggestion and hope that these additional data address your concerns.

5. The authors need to explain how to decide the condition of stirring culture with microcarriers (e.g., rotation speed, cell seeding density). The condition was optimized specifically for the cells produced in this study? Is it the same as the method reported previously (references 18, 20)? Which part was developed by the authors? Again, the cells at P10 and P40 show the similar behavior in this experiment?

Thank you for your insightful question and the opportunity to further explain our experimental design.

Our method of stirring culture with microcarriers indeed differs from the approaches described in references 18 and 20.

	Ref.18	Ref.20	Our method
Cell density	4 × 10 ⁴ cells/mL	1.5 × 10 ³ cells/cm ²	5 × 10 ⁴ cells/mL
volume	120mL	30mL (10 cm ² /mL)	50mL
Spin speed in first 24h	50rpm	60rpm	30rpm 5mins, sit for 1h, repeat for 24 times
Spin speed afterwards	50rpm	60rpm	40rpm

Culture time to reach the max cell number	5 days	110 h (4.58 days)	6 days
--	--------	-------------------	--------

The conditions we used were primarily derived from the microcarrier manufacturer's recommendations for fibroblast cells. Given the consistent results and high adaptability of our Pig#4 cells to these conditions (Fig.5), we didn't find further optimization necessary at this stage of our research. This adaptability also suggests that Pig#4 cells have a high level of culture adaptability, which is a valuable characteristic for cell-cultured meat production. In this study, our primary aim was to demonstrate that Pig#4 retains the ability to proliferate on microcarriers and maintain adipogenic capability, similar to primary seed cells.

Regarding your final question about the behavior of cells at P10 and P40, we have indeed verified their growth performance on edible microcarriers. Our results showed that the growth performance of the cells remained consistent, which underscores the robustness of Pig#4 cells in these culture conditions (Fig.5C).

I hope this response provides further clarity on our methodology and results. We appreciate your thoughtful questions as they enable us to present our work more effectively.

6. As the authors described in the manuscript, C2C12 is mouse muscle cell line. In this study, on the other hand, adipocytes were produced from an appropriate animal source for cultured meat production, and the authors claimed the advantage of this study. C2C12 have been widely used in a large number of previous studies, since it is easy to induce myotube differentiation. In addition, the mouse muscle cells and porcine fat cells were just co-cultured. Although the both cells showed differentiation into myotubes and adipocytes in the culture dish, they didn't show the marbling pattern. This co-cultured cell assembly was not similar to real meat consisting of myofibers and fat components.

Thank you for your insightful comments. We agree with your points and have revised our manuscript accordingly.

In light of your comments, we have revised our manuscript to remove the claim about the marbling pattern. This study primarily aims to establish a method for obtaining highly adipogenic seed cells, and the co-culture system serves as a preliminary demonstration of their potential application in cell-cultured meat production for cost reduction.

7. Unpublished data should be not used to discuss the study since the readers cannot judge the reliability of the statement.

Thank you for bringing this to our attention. We appreciate your feedback and understand the importance of referring only to published and verifiable data for discussion in our study.

We have revised the manuscript and removed the section references to unpublished data. We are grateful for your careful review and helpful suggestions.

Reviewer #2:

Overall:

This study is certainly in an important area, and the science presented is generally of good quality, for which I would commend the authors.

However, whilst it certainly is an interesting observation that an immortalised porcine cell line retained adipogenic differentiation potential for many generations, this is not an especially profound finding, and the additional characterisation of this observation presented here, is (in this reviewer's opinion) not sufficient to merit publication in a journal such as Communications Biology. Several significant further elements would need to be added, in this reviewer's opinion, to demonstrate the potential of this cell line for use in cultured meat research and production bioprocesses.

Response: Thank you for your constructive feedback and appreciation of the important area our study addresses. We understand your concern about the depth of our findings and their suitability for a prestigious publication like Communications Biology.

The main goal of our study was to exploit Pig#4, a characterized immortalized porcine cell strain retaining adipogenic differentiation over many generations, to explore its application potential in the cultured meat field. While we acknowledge that this might not seem profound in isolation, we believe it represents a significant advancement in the field of cell-cultured meat research for a few reasons:

- a) Creating and characterizing a stable porcine cell line that can be reliably used to produce adipocytes could significantly reduce the costs and logistical challenges associated with primary cell isolation and culture, which is currently a major bottleneck in the field.
- b) While there are instances of using immortalized chicken cells in cell-cultured fat studies, it's important to emphasize that poultry preadipocyte cells have limited ability to synthesize lipids (6), with lipid accumulation being primarily attributable to the supplementary oleic acid in the culture medium. Our study acts as a proof-of-concept for the utilization of immortalized mammalian cells in cell-cultured fat production, potentially laying the groundwork for the development of additional seed cell lines from other farm animals.

- c) Significantly, this study underscores the potential of employing clonal selection to augment the performance of seed cells. Through this approach, we succeeded in developing seed cell strains with capabilities not found in primary seed cells, such as in-situ differentiation on microcarriers or co-differentiation with myogenic cells.
- d) We have successfully isolated the Pig#4 cell strain and demonstrated its versatility across various cell-cultured meat applications. In doing so, we have furnished researchers with a readily accessible experimental model that is well-placed to make significant contributions to the advancement of the cell-cultured meat industry.

However, we take your feedback to heart and agree that additional studies could further demonstrate the potential of this cell line for use in cultured meat production bioprocesses. We are currently working on additional experiments to further characterize this cell line, including its behavior in bioreactor-based culture systems and its ability to form structured adipose tissue in co-culture with muscle cells.

Major Comments:

1. The use of many animal derived products, principally FBS in the medium (for both proliferation and differentiation), and porcine gelatin-derived microcarriers, are incompatible with accepted philosophies for cultured meat production. Whilst the presented results around upscaled cultures are promising, they do not yet represent an actual proof-of-principle for this cell line. Various defined media formulations have been developed (and are in the public domain), at the very least a selection of these should be tested in short-term proliferation assays.

Thank you for your thoughtful comments and suggestions.

We understand and agree with your concerns about the use of animal-derived products, primarily Fetal Bovine Serum (FBS), in the medium, and porcine gelatin-derived microcarriers. These may not align with the widely accepted principles of cultured meat production, where the goal is to minimize the reliance on animal-derived products.

In line with this, we have strived to minimize serum use (as shown in Fig. 2 for differentiation in 2% FBS), and use peanut wire-drawing protein scaffold to amplify Pig#4 (SupFig.4). As you kindly suggested, we further explored the possibility of culturing Pig#4 in a commercially available serum-free medium (Serum-free Medium for Adipose-derived MSC; NC0103+NC0104.S; Yocon). Interestingly, we observed that Pig#4 exhibited significantly faster growth in serum-free medium than in DMEM with 10% FBS during the initial 48h. However, we observed cell death and cessation of growth at 72h.

These findings suggest that further optimization of the serum-free medium for Pig#4 may be necessary, perhaps through the addition of more nutrients or growth factors which currently unclear.

Again, thank you for your valuable insights, which will undoubtedly contribute to the improvement of our work.

Respond Fig.2 Growth arrest in serum-free medium of Pig#4.

20,000 cells were cultured in a 35mm dish with either serum-free medium or DMEM+10%FBS. Images were taken with a 4x objective lens on a bright-field microscope.

2. Whilst the acceptability of genetic modifications of different types is rather unclear for cultured meat, and will certainly vary between jurisdictions, the method of immortalisation presented here (use of an SV40 large T antigen) is amongst the most contentious, given that it uses viral transgenic material. The authors do allude to other possibilities for immortalisation in their discussion, but in my opinion, some actual data is needed here. It seems likely that a combination of telomerase and CDK4, or telomerase together with knockout of p16, might be sufficient for immortalisation.

Thank you for your insightful comments and constructive criticism. We appreciate and share your concern regarding the use of SV40 large T antigen for cell immortalization, given the contentious nature of using viral transgenic material.

However, the primary aim of our research was to validate the concept that immortalized cells could serve as seed cells for cell-cultured meat, demonstrating superior performance compared to primary seed cells. We believe this concept, regardless of the specific method used for immortalization in this research, represents an important step towards the development of cell-cultured meat. Furthermore, we hope our work with Pig#4 will serve as a valuable model for this field, thanks to its cost-effective maintenance and high adipogenesis efficiency.

We agree with your suggestion of using a combination of telomerase and CDK4 or telomerase and knockout of P16 as potentially less contentious methods of immortalization. Indeed, we have been exploring alternative methods to generate immortalized adipose precursor cells. We have established a novel porcine preadipose cell strain, Swine#2C10, through spontaneous immortalization and unlimited dilution. This cell strain exhibits comparable differentiation capabilities to Pig#4 (Respond Fig.3), though with a significantly slower growth rate (data not shown). We are currently conducting further research on this cell strain and plan to present these findings in a future publication.

Again, thank you for your valuable input, which will undoubtedly contribute to the advancement of our work.

Respond Fig.3 Swine#2C10 exhibit a similar adipogenesis capacity to Pig#4

A. Oil-Red staining of differentiated Swine#2C10 and Pig#4.

B. Oil-Red was extracted with isopropanol and measured at 500nm.

3. The use of rodent C2C12 cells, which are known to differentiate extremely robustly, for the co-differentiation experiments, is unfortunately not a particularly impressive proof-of-principle when it comes to co-culture. Porcine primary cells (even if early passage, not immortalised) would have been much more informative.

Thank you for your insightful comment. You're correct that using rodent C2C12 cells, known for their robust differentiation, is not the most compelling proof of the principle for co-culture. Our limited experience with primary muscle cell culture and differentiation primarily drove our selection of C2C12.

In response to your suggestion, we have managed to obtain immortalized porcine muscle satellite cells and co-cultured them with Pig#4 (SupFig.8). The results were similar to those obtained with the C2C12 co-induction, showing that Pig#4 can co-grow and co-mature with porcine muscle cells, producing a pattern similar to intramuscular fat. These findings reinforce the value and relevance of our Pig#4 model, and we appreciate your comment which prompted this additional line of investigation.

4. The adipogenic differentiation media contains several substances of concern (as noted by the authors in their discussion). Porcine FAP differentiation has

recently been demonstrated with a simplified formulation (Mitic et al., 2023), which the authors should consider testing.

Thank you for your thoughtful suggestion. We appreciate the reference to the simplified formulation for porcine FAP differentiation proposed by Mitic et al. (2023). We, too, thought the formulation to be ingeniously simple and were hopeful about its potential.

Following your recommendation, we prepared the differentiation medium as per the formulation detailed in the referenced article and used it to induce Pig#4. Unfortunately, this medium did not prove suitable for Pig#4, as we observed cell detachment and clustering after 3 days of incubation with this medium(Respond Fig.4).

One possible explanation for this outcome could be that the cells used in the study by Mitic et al. were primary cells, whereas Pig#4 is a cell strain with exceptionally high adipogenesis capacity. Similar to our experience with serum-free mediums, it appears that Pig#4 may require specific nutrients or growth factors not included in the simplified formulation. We believe further optimization of the differentiation medium would be necessary for successful differentiation of Pig#4.

Respond Fig.4 Pig#4 could not differentiate with serum-free medium.

Pig#4 was cultured in 60mm dishes until confluent, and the medium was changed into ADM or serum-free differentiation medium, as Mitic et al., 2023 describe. The medium was changed every 48h. Images were taken with a 10x objective lens on a bright-field microscope daily.

5. Figure 2: The 2x5 and 4+4 medium strategies are hard to compare as they have different amounts of time in growth and differentiation medium. Better controls would have been good to see, but I suppose the authors are comparing the medium strategy and not doing a direct comparison of the two differentiation media. I would have also liked to see a negative control without inducers, which

would have helped to show if 2x5 is mostly just causing 'passive' lipid accumulation or true adipogenesis.

Thank you for your insightful comments. We acknowledge that comparing the 2x5 and 4+4 medium strategies, given their different time frames in growth and differentiation medium, may not provide a straightforward comparison. Upon reflection, we agree that this comparison may lead to confusion and have thus removed it from the manuscript.

In response to your suggestion regarding a negative control for the 2x5 strategy, we have conducted further experiments. These experiments included a control group without insulin to evaluate whether the 2x5 strategy primarily induces 'passive' lipid accumulation or true adipogenesis. The results of this analysis have been included in Supplementary Figure 3.

Minor Comments:

6. Line 223. The resulting Pig#4 cultured fat dramatically differed from the empty alginate hydrogel in appearance, the yellow hue of Pig#4 cultured fat is similar to other cultured fat with primary cells reported before (Fig. 3B).

It would be interesting here to see a comparison with undifferentiated cells.

Thank you for your insightful observation and the compelling question you've raised. Indeed, the difference between alginate hydrogels with and without cells is rather subtle. The hydrogel with undifferentiated cells appears slightly less transparent due to the presence of many Pig#4 cells, which obstruct the passage of light. However, as undifferentiated Pig#4 cells are "transparent", the obstruction is minimal. For further clarification, please refer to the attached figure, which provides a comparison before and after induction. The image on the left depicts the alginate hydrogel with cells.

Respond Fig. 5

Photograph of alginate hydrogel with Pig#4 before (left) and after differentiation (right), scale=5 mm.

7. Line 230. Interestingly, most cavities in the alginate were occupied by 2-3 cells, indicating that Pig#4 may divide in the alginate hydrogels (Fig. 3C).

This seems rather speculative, the authors might consider EdU staining or try DNA quantification to explore this further.

We appreciate your insightful suggestion. To investigate this further, we have performed Ki67 immunofluorescence staining on alginate sections. We noted a strong proliferative activity in Pig#4 cells on day 0, with a majority of them being Ki-67 positive. These findings corroborate our hypothesis that the observation of multiple cells within each cavity is likely due to cell proliferation within the alginate hydrogel during Day-2 to Day0.

8. Figure 4: The 10% FBS 4+4 confocal image is horribly overexposed and needs to be retaken.

Thank you for bringing this to our attention. We apologize for the oversight. The overexposed image in Figure 4 has been replaced with a more appropriately exposed one.

9. The statistical analysis and data presentation is generally sound, and the descriptions of the methodology are sufficiently detailed.

Thank you for your kind words and positive feedback on our statistical analysis, data presentation, and methodology descriptions. We greatly appreciate your careful review and thoughtful comments.

Reviewer #3 (Remarks to the Author):

Title: An immortal porcine preadipose cell strain for efficient production of cell-cultured fat

The focus of the manuscript is the development of immortalized pre-adipose cells and the optimal culture conditions for the production of cultured meat. The manuscript discusses several aspects related to pre-adipose cell line Pig#4, including its potential for differentiation, the optimal adipogenic method, its proliferation and differentiation in 3D culture, and its co-differentiation with rodent myoblasts (C2C12). This manuscript is commendably well organized and written, with a conclusion that appropriately derives from the experimental data. However, it is important to acknowledge the limitations of this study, such as the use of immortalized cells and rodent cells for co-differentiation. Despite these limitations, the study's overall impact on cell culture technique and the utilization of edible microcarriers to reduce the cost of FBS is undeniably impressive. Nonetheless, there are a few concerns and suggestions that could enhance the manuscript further. However, there are some concerns and suggestions for improving the manuscript.

Response: Thank you for your thoughtful and constructive feedback. We greatly value your recognition of our manuscript's organization and the findings it presents.

We acknowledge the limitations associated with using immortalized and rodent cells for co-differentiation, as you correctly pointed out. We agree that the incorporation of

porcine muscle cells would indeed provide a more realistic model. To this end, we have included data in the revised manuscript that demonstrates our efforts in this direction.

Your commendation regarding our utilization of edible microcarriers and the reduction of FBS costs is greatly appreciated. We agree that these aspects are crucial to making cultured meat a viable and more sustainable alternative to conventional meat production.

We believe your expert perspective will greatly aid in refining our work and will guide our future research in this exciting field. Once again, thank you for your time and consideration.

Major concerns

1. Please consider including a discussion on the limitations of this study and proposing future directions for more meaningful, edible, and cost-effective cell-cultured fat. This would provide valuable insights for further research.

Thank you for your insightful comments. We have revised the discussion section to include an examination of the study's limitations and potential future developments. We believe this additional information will help guide further research in this field.

2. The 2x5 protocol is an impressive method as it avoids the use of potentially toxic differentiating agents. However, the mRNA expressions of the 2x5 protocol are higher than those of the 4+4 protocol in Fig. 2D. In addition, in Fig. 2C, the lipid accumulation of the 4+4 protocol is higher than that of the 2x5 protocol, but the mRNA expressions of adipose-specific genes (FABP4, PLIN1, and AdipoQ) are lower. Please explain the difference between the fat accumulation data and the gene expression data.

Thank you for pointing out the discrepancy between the lipid accumulation data in Fig. 2C and the mRNA expression data in Fig. 2D. As Reviewer2 noted, it is not appropriate to make direct comparisons in this manner. Therefore, in response to your insightful comment, we have decided to remove Fig. 2D from the manuscript. We appreciate your attention to detail and your helpful suggestions for improving the clarity and accuracy of our report.

3. In Fig. 6D, the mRNA expressions of adipose-specific genes (FABP4, PLIN1, and AdipoQ) increased approximately two-fold on day 12 compared to day 8. The expressions of adipose-specific genes decreased when cells were treated with adipogenic maintenance medium (AMM) between days 4 and 8, but increased between days 8 and 12 when AMM (with 2% horse serum) was treated. The authors should provide a more comprehensive explanation regarding the increase of adipose-specific genes on day 12.

Thank you for your keen observation and insightful suggestion. Our hypothesis for the observed decrease in adipose-specific gene expression from day 4 to day 8 is that the rosiglitazone present in the adipogenic differentiation medium (ADM) boosts the

expression of fat-related genes on day 4. Conversely, the adipogenic maintenance medium (AMM), which contains only insulin, might account for the significant drop in these genes expression levels from day 5 to day 8.

As the culture period in AMM extends (regardless of whether it contains 2% FBS or 2% HS), the cells gradually accumulate more lipids. As evidenced in Fig. 2A, the lipid droplets at day 12 are larger than at any previous time point, which likely explains the increase in fat-related gene expression from day 8 to day 12.

We greatly appreciate your insightful comment, which has encouraged a deeper analysis of our results.

4. To utilize in cultured meat production, it is crucial that the cells adhere well to the 3D structure. However, in the textured soy protein (TSP) used in the manuscript, the cell adhesion seems to be inadequate. Is there an experiment to evaluate the adhesion ability of Pig#4 and other cells? Alternatively, have the authors explored any methods to enhance the adhesion ability?

Thank you for raising this important question. We suspect that the apparent lack of cell adhesion to the TSP, as observed in our manuscript, is attributable to the process of paraffin sectioning that we employed. Under normal culture conditions, the Pig#4 cells typically adhere closely to the TSP, which is evidenced by the absence of any observed cell clumps during the culture period.

However, the process of dehydration, which is necessary for paraffin sectioning, can cause different degrees of shrinkage in the cell clumps and the TSP. We believe this differential shrinkage is responsible for the apparent detachment of cells from the TSP observed in the sections. We appreciate your insightful query and will consider further experiments to evaluate the adhesion ability of Pig#4 and other cells to the TSP.

Minor concerns

5. The authors should standardize the gene abbreviations by choosing either uppercase or lowercase letters and using them consistently throughout the figures.

We apologize for the inconsistency in the gene abbreviations across the figures. We have now standardized the gene abbreviations in Fig.7 and ensured they are consistent with those in the other figures. Thank you for bringing this to our attention.

6. Keywords contain a mix of capital and lowercase letters.

We apologize for this oversight and have now corrected it to ensure consistent use of lowercase letters. We appreciate your attention to detail.

7.L67: Wewe

Thank you for catching this error. We've corrected the capitalization issue on line 67 as you suggested. We appreciate your meticulous attention to detail.

References

1. Peng X, Song T, Hu X, Zhou Y, Wei H, Peng J, et al. Phenotypic and Functional Properties of Porcine Dedifferentiated Fat Cells during the Long-Term Culture In Vitro. *Biomed Res Int.* 2015;2015:673651.
2. Nobusue H, Kano K. Establishment and characteristics of porcine preadipocyte cell lines derived from mature adipocytes. *Journal of Cellular Biochemistry.* 2009;n/a-n/a.
3. Yuen Jr JSK, Saad MK, Xiang N, Barrick BM, DiCindio H, Li C, et al. Aggregating in vitro-grown adipocytes to produce macroscale cell-cultured fat tissue with tunable lipid compositions for food applications. *eLife.* 2023;12.
4. Dohmen RGJ, Hubalek S, Melke J, Messmer T, Cantoni F, Mei A, et al. Muscle-derived fibro-adipogenic progenitor cells for production of cultured bovine adipose tissue. *NPJ Sci Food.* 2022;6(1):6.
5. Mitić R, Cantoni F, Börlin CS, Post MJ, Jackisch L. A simplified and defined serum-free medium for cultivating fat across species. *iScience.* 2023;26(1).
6. Nematbakhsh S, Pei Pei C, Selamat J, Nordin N, Idris LH, Abdull Razis AF. Molecular Regulation of Lipogenesis, Adipogenesis and Fat Deposition in Chicken. *Genes.* 2021;12(3).

Reviewers' comments:

Reviewer #1 (Remarks to the Author):

This manuscript have been partly revised to respond the comments of reviewers. However, the revisions are still not sufficient to be published in the Journal.

Reviewer #2 (Remarks to the Author):

Overall the revised manuscript represents a major improvement on the initial submission, and although I retain significant reservations about the relevance of an SV40-immortalised cell line for cultured/cultivated meat applications, I believe this work could now be considered for publication in Communications Biology, pending a further round of revisions, for which a small amount of experimental work is still required.

Major Comments:

- The muscle differentiation characterisation presented in the co-culture experiments, both with C2C12 and porcine myoblasts, is limited to RNA (plus an F-actin staining, which is not muscle specific and not remotely convincing). Some muscle protein-specific readouts, either with western blot or immunofluorescence, is required to augment this section of the manuscript and provide conclusive evidence of muscle differentiation at the protein level. Significant evidence of myofusion is not visually evident in the brightfield images, in my opinion (maybe some of the images contain 2-3 small myotubes), which makes me somewhat suspicious regarding the level of muscle differentiation achieved in these co-culture experiments.
- The discussion section does contain a short mention of the SV40 nature of the immortalization strategy employed here, but it is very brief. This needs to be lengthened to emphasise the problems with SV40-based immortalisation, include some discussion of alternative immortalisation methods (both using genetic engineering and spontaneous immortalisation), and perhaps some of the other points mentioned by the authors in their rebuttal document.

Minor Comments:

- the manuscript requires a very thorough proofreading and spell-checking, as it contains many typographical errors, both in the text and in the figures, some of which are quite serious (e.g. 'MOYG' is listed as a gene name). In some cases entire parts of figure legends are also missing (e.g. Figure 1G). There are too many examples of such errors for me to individually list them all here.
- I would advise the authors to consider changing the name of the cell name to something more specific with respect to it's origin or it's differentiation/tissue capacity, or both.

Reviewer #3 (Remarks to the Author):

The authors responded effectively to the reviewer's comments, and the current version of the manuscript meets the necessary standards.

Responses to Reviewers' comments

Reviewer #2 (Remarks to the Author):

Overall the revised manuscript represents a major improvement on the initial submission, and although I retain significant reservations about the relevance of an SV40-immortalised cell line for cultured/cultivated meat applications, I believe this work could now be considered for publication in Communications Biology, pending a further round of revisions, for which a small amount of experimental work is still required.

Response: We appreciate the time and effort you have invested in evaluating our manuscript. We are heartened to know that our work has significantly improved with your insightful feedback. It is encouraging to hear that our revised manuscript is now under consideration for publication in Communications Biology. This acknowledgement not only validates our efforts, but also motivates us to continue refining our work in response to your valuable suggestions.

Major Comments:

1. The muscle differentiation characterisation presented in the co-culture experiments, both with C2C12 and porcine myoblasts, is limited to RNA (plus an F-actin staining, which is not muscle specific and not remotely convincing). Some muscle protein-specific readouts, either with western blot or immunofluorescence, is required to augment this section of the manuscript and provide conclusive evidence of muscle differentiation at the protein level. Significant evidence of myofusion is not visually evident in the brightfield images, in my opinion (maybe some of the images contain 2-3 small myotubes), which makes me somewhat suspicious regarding the level of muscle differentiation achieved in these co-culture experiments.

Response: Thank you for your insightful comments and constructive suggestions. We agree that the muscle differentiation characterization in our co-culture experiments needs further validation.

In response to your feedback, we have conducted additional experiments using specific antibodies to investigate the differentiation status of the cells co-cultured with ISP-4 and C2C12. The results show a clear presence of myotubes, providing evidence of muscle differentiation at the protein level. These new data have replaced the previous F-Actin staining in the revised manuscript (Fig. 7C).

For the co-culture of ISP-4 and porcine myoblasts, we also attempted the same antibody staining. However, as the antibody does not target pig proteins, we did not detect any signal. Following your suggestion, we employed western blot to monitor the muscle marker protein during the differentiation process. While we did not get any MYL1 signal, we observed a gradual increase in DESMIN upon induction, indicating the occurrence of muscle

differentiation.

We believe these additional experiments have addressed your concerns and provide more robust evidence for muscle differentiation in our co-culture system. We hope these changes will strengthen our manuscript and look forward to hearing further feedback.

2. The discussion section does contain a short mention of the SV40 nature of the immortalization strategy employed here, but it is very brief. This needs to be lengthened to emphasise the problems with SV40-based immortalisation, include some discussion of alternative immortalisation methods (both using genetic engineering and spontaneous immortalisation), and perhaps some of the other points mentioned by the authors in their rebuttal document.

Response: Thank you for your insightful comments. We appreciate your suggestion to further emphasize the discussion on the SV40-based immortalization strategy and its alternatives in our manuscript.

In response to your feedback, we have expanded the discussion section, specifically paragraphs 3, 4, and 5. In these paragraphs, we have now included more detailed discussion on three different methods of inducing cell immortality. These methods include SV40-based immortalization, Cas9-based genetic engineering strategies, and spontaneous immortalization. Furthermore, we have also elaborated on their potential applications in the context of cell-cultured meat seed cells.

We believe these additions will provide a more comprehensive picture of the current state of the field, as well as the challenges and opportunities associated with each immortalization strategy.

Once again, we thank you for your constructive feedback and are committed to further improving our manuscript in light of your suggestions.

Minor Comments:

1. the manuscript requires a very thorough proofreading and spell-checking, as it contains many typographical errors, both in the text and in the figures, some of which are quite serious (e.g. 'MOYG' is listed as a gene name). In some cases entire parts of figure legends are also missing (e.g. Figure 1G). There are too many examples of such errors for me to individually list them all here.

Response: We sincerely appreciate your attention to detail and your feedback on this matter. We apologize for the oversights in the manuscript and figure legends, including the typographical errors and missing parts.

We take this feedback seriously and understand the importance of maintaining high standards of accuracy in our work. We have conducted a thorough proofreading and spell-checking of the entire manuscript and have made corrections wherever necessary. We have also revised all the figure legends for completeness and accuracy, ensuring that they accurately represent the corresponding figures.

2. I would advise the authors to consider changing the name of the cell name to something more specific with respect to its origin or its differentiation/tissue capacity, or both.

Response: Thank you for your insightful suggestion. We agree that a more specific name would effectively reflect the origin and differentiation capacity of the cell line.

In light of your advice, we have decided to rename the "Pig#4" cell line as "ISP-4", which stands for Immortalized Swine Preadipocytes #4. We believe this name provides a clear indication of the cell strain's origin and its potential for differentiation. We have updated the manuscript accordingly to reflect this change.